# The rise of predation in Jurassic lampreys

Feixiang Wu [1] ✉, Philippe Janvier[2] & Chi Zhang [1] ✉

Lampreys, one of two living lineages of jawless vertebrates, are always intriguing for their feeding behavior via the toothed suctorial disc and life cycle comprising the ammocoete, metamorphic, and adult stages. However, they left a meager fossil record, and their evolutionary history remains elusive. Here we report two superbly preserved large lampreys from the Middle-Late Jurassic Yanliao Biota of North China and update the interpretations of the evolution of the feeding apparatus, the life cycle, and the historic biogeography of the group. These fossil lampreys' extensively toothed feeding apparatus differs radically from that of their Paleozoic kin but surprisingly resembles the Southern Hemisphere pouched lamprey, which foreshadows an ancestral flesh-eating habit for modern lampreys. Based on the revised petromyzontiform timetree, we argued that modern lampreys' three-staged life cycle might not be established until the Jurassic when they evolved enhanced feeding structures, increased body size and encountered more penetrable host groups. Our study also places modern lampreys' origin in the Southern Hemisphere of the Late Cretaceous, followed by an early Cenozoic antitropical disjunction in distribution, hence challenging the conventional wisdom of their biogeographical pattern arising from a post-Cretaceous origin in the Northern Hemisphere or the Pangean fragmentation in the Early Mesozoic.

As a lineage of the living jawless vertebrates, lampreys have great weight in the study of vertebrate evolution[1–5]. They are characterized by their peculiar feeding behavior of eating blood or cutting off tissues from the hosts or prey to which they firmly attach via their toothed oral sucker[6,7]. In such a way, lampreys play a significant role in the aquatic ecosystem and, in some cases, where they are non-native, even bring tremendous loss to the fishery economy[4]. These jawless fishes have been in existence for ca. 360 million years but left an extremely patchy fossil record in the post-Carboniferous period, with only two species known from the Cretaceous[2,3,8,9]. Despite the superficially conservative morphology throughout their history, from the simply assembled teeth in Paleozoic fossils, lampreys' feeding apparatus, especially the size, shape and arrangement of the keratinous teeth, was substantially reformed and enhanced to the pattern of modern species[2,3,6,7,9–12]. And evidently, departing from their Paleozoic kin with non-ammocoete larvae and expanding the habitats to the freshwater domain, lampreys changed the life-history strategy some time before the Cretaceous by

evolving the ammocoete and metamorphic stages[3,9,13–15]. Eventually, they established their current diversity and anti-tropical distribution[5,16]. However, these scenarios were poorly understood due to the lack of reliable fossil record and the disputed lamprey phylogeny, especially that of the crown-group lineages[1–5,16]. Here we shed light on these issues by reporting two lampreys from the Jurassic terrestrial fossil Lagerstätte Yanliao Biota of North China[17]. These fossil lampreys were exquisitely preserved with a complete suite of feeding structures, including the well-developed movable biting plates on the tongue-like piston, which has never been clearly recognized in previously known fossil lampreys and astonishingly resembles the pouched lamprey (*Geotria australis*[12]) now confined to the Southern Hemisphere. Bridging the recorded fossil and extant lampreys, these fossils offer an opportunity to reconstruct the evolutionary process and the ancestral state of modern lampreys' feeding biology. As key constraints of the evolutionary timeline of its group, they also make it possible to assess the coevolutionary interactions with the potential

[1]Key Laboratory of Vertebrate Evolution and Human Origins of Chinese Academy of Sciences, Institute of VertebratePaleontology and Paleoanthropology, Chinese Academy of Sciences, 100044 Beijing, China. [2]Muséum national d'Histoire naturelle, UMR 7207, CP38, 8, rue Buffon 75231, Paris Cedex 05, France. ✉e-mail: wufeixiang@ivpp.ac.cn; zhangchi@ivpp.ac.cn

hosts (or prey) and their implications for the establishment of the modern-type life-history mode. Also, based on the revised lamprey phylogeny, the early biogeographical history of modern lampreys was reconstructed and their 'poles-apart' distribution pattern was reinterpreted.

## Results

### Systematic paleontology

**Order.** Petromyzontiformes Berg, 1940[18]

**Genus _Yanliaomyzon_. gen. nov.**

**Type species.** _Yanliaomyzon occisor_ gen. et sp. nov.

**Diagnosis.** Stem lampreys with oral discs well-toothed in anterior and lateral fields; anterior and lateral oral disc teeth closely arranged, dorsally truncated, spatulate in shape with the slightly concaved undersurface of the free edge protruding a shallow blade; posterior disc teeth lacking, anterior and lateral circumoral teeth elongate and trihedral in shape; supraoral lamina large and consisting of two stout central cusps flanked by wing-like lateral extensions; transverse lingual lamina very large with the apices of three cusps interlocking with the supraoral lamina in vivo.

**Included species.** _Yanliaomyzon ingensdentes_ gen. et sp. nov.

**Etymology.** 'Yanliao' derives from Yanliao Biota, a Jurassic terrestrial Lagerstätte from North China[17], where these fossils were discovered; 'myzon' (Greek), sucker.

**_Yanliaomyzon occisor_. gen. et sp. nov. (Figs. 1a–e, 2e, f, Supplementary Figs. 1g–i, 2a–g, k, l)**

**Etymology.** Latin _'occisor'_, meaning 'killer', refers to the powerful hunting skill of the species.

**Holotype.** IVPP V 15830, a completely preserved lamprey.

**Paratype.** IVPP V 18956A, B, a lamprey with the head and anterior trunk preserved.

**Horizon and locality.** Tiaojishan Formation, Oxfordian, earliest Late Jurassic, ca. 158.58–160 million years ago (Ma)[19]; Daxishan, Toudaoyingzi Town Jianchang County, Liaoning Province (Holotype), and Nanshimen Village, Gangou Town, Qinglong County, Hebei Province (Paratype), China.

**Diagnosis.** The supraoral lamina spanning completely the lateral rims of the oral aperture, with the central cusps flanked immediately by two smaller projections; 16 circumoral teeth; the tail region occupying slightly less than 28% of the total body length.

**_Yanliaomyzon ingensdentes_. gen. et sp. nov. (Figs. 1f–h, 2a–d, e, Supplementary Figs. 1a–f, 2h–j)**

**Etymology**
Latin _'ingens + dentes'_, meaning large teeth, refers to the large cuspid laminae on the gouging piston.

**Holotype.** IVPP V16715A, B, a completely preserved individual.

**Paratype**
IVPP V 16716A, B, an exquisitely preserved toothed oral disc and the laminae on the tongue-like piston.

**Horizon and locality.** Daohugou beds, Callovian, late Middle Jurassic, ca. 163 Ma[19] in Wubaiding Village, Reshuitang County, Liaoning Province, China.

**Diagnosis.** The supraoral lamina occupying roughly one-third of the rim of the oral aperture; the transverse lingual lamina almost equaling to the supraoral lamina in width; ca. 23 circumoral teeth; the tail region occupying slightly more than 40% of the total body length.

### Description and comparison

_Yanliaomyzon occisor_ and _Y. ingensdentes_ are fairly large (Supplementary Table 1) with the former one (642 mm) being among the largest of the group, only smaller than the anadromous sea lamprey _Petromyzon marinus_ (maximum adult length 1200 mm, same for below), Pacific lamprey _Entosphenus tridentatus_ (850 mm), pouched lamprey _Geotria australis_ (788 mm) and Arctic lamprey _Lethenteron camtschaticum_ (790 mm)[7,12].

The most conspicuous features of _Yanliaomyzon_ species are the extensively toothed oral disc (od) and tongue-like piston (pt) (Supplementary Note 1, Figs. 1, 2a–f, Supplementary Figs. 1b–d, h, i, 2h–l). Their dentitional arrangement, similar to one another in general morphology, strikingly resembles that of pouched lamprey _Geotria australis_ (Fig. 2g) currently distributed in the Southern Hemisphere[10,11]. The oral disc occupies 5.2% of the total body length in _Y. occisor_, which falls within the range (2.7–6.3%) of the living flesh-eating lampreys[11]. The size of the oral disc cannot be accurately measured for _Y. ingensdentes_ because this part was distorted during the preservation. In the anterior and lateral fields of the oral disc, the closely arranged disc teeth (dt) are dorsally flattened, slightly concave in the posterior surface with the blade-like dorsal edge pointing toward the center of the oral aperture. The morphology of these teeth, together with their absence in the posterior field, is reminiscent of the pattern in pouched lamprey _Geotria australis_[10,11,20], although they are less numerous than in _Geotria_[20]. The circumoral teeth (cot) of _Yanliaomyzon occisor_ and _Y. ingensdentes_ are similar in general morphology but those in the former are proportionally larger and less numerous than in the latter. These teeth are much larger than other oral disc teeth and trihedral in shape, with the longest of the three cutting edges facing slightly sideways and posteriorly (Fig. 2a–f). By analogy, with such an arrangement, these oral disc teeth might function similarly to their counterparts of _Geotria australis_ in allowing the forward sliding of the oral disc over the prey's body surface and resisting any tendency to slip backward or laterally of the disc, thereby coordinating the biting movement of the transverse lingual lamina (tl) on the tongue-like piston cartilage[10,11]. The posterior circumoral teeth are lacking in _Yanliaomyzon_ species as in _Lampetra_ species[11,12] and _Geotria macrostoma_[19], whereas they are usually the largest teeth in the complete circumoral series in _Geotria australis_[10,20]. The supraoral laminae (so) are very large and all have a pair of sharp central inner cusps flanked by a pair of large and curved wings as in _Geotria_[10,20]. However, this lamina of _Yanliaomyzon occisor_ differs from that of _Y. ingendentes_ in its longer lateral extension and two accessory projections lateral to the central cusps (Figs. 1b, f, g, 2a–d, f, Supplementary Information). The wide infraoral laminae (io) of both species are surmounted by a straight row of about nine to ten stout cusps. The transverse lingual laminae are remarkably large and occupy most of the part of the oral aperture (Fig. 2a–f, Supplementary Figs. 1g–i, 2h–j). This lamina is thus proportionally much larger than its counterpart in living lampreys[10–12,20], whereas this structure is likely much less developed, if present, in all other known fossil lampreys[3,8,9,21–27]. Despite the variations in the relative size and the shape of the ventral part, the general morphology of this key feeding structure is similar in both _Yanliaomyzon_ species. This toothed lamina is convexly curved and characterized by a prominent central cusp flanked by two slightly smaller lateral cusps like in the feeding adults of _Geotria australis_[10,11].

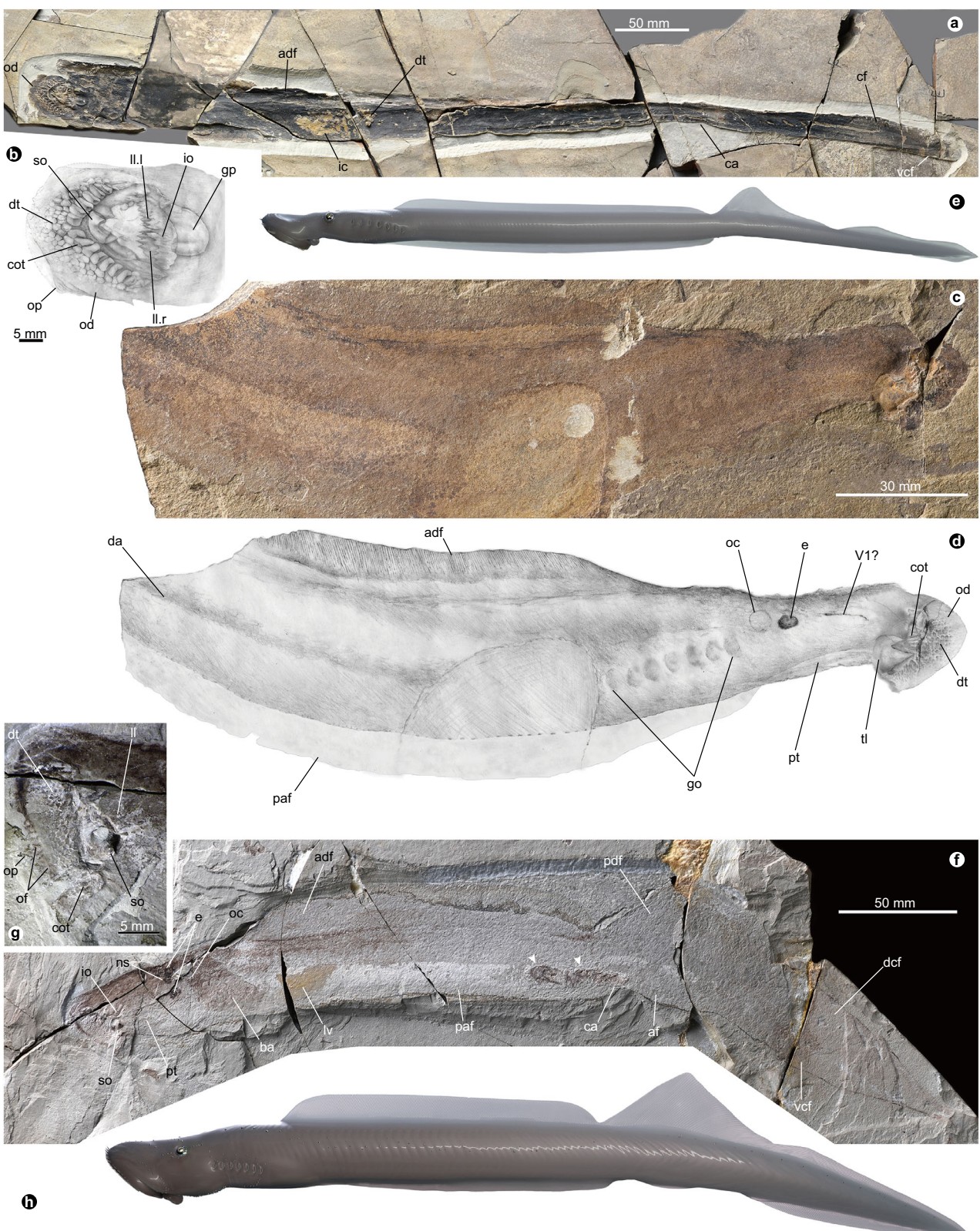

The tips of these three cusps, when raised upward in life, interdigitate with the grooves on the rear side of the supraoral lamina (Fig. 2a–f), which, again, recalls the feature unique to Geotria australis among living lampreys and allows powerful biting and removing large amounts of flesh from the prey[10,11]. The cusps of the longitudinal lingual laminae (ll, ll.l, ll.r) are strong and closely arranged, making this lamina an extensively serrated cutting edge (Figs. 1b, 2a–d).

The prebranchial region is longer than the branchial apparatus (ba) (Fig. 1c–e, f, h), of which the range can be roughly determined by the positions of the branchial pouches (bp) and external gill openings (go). The eyes (e) are medium-sized and oval in shape, with the otic capsules (oc) positioned immediately behind (Fig. 1c, d, f). The gular pouch (gp) of these lampreys (Figs. 1b, e, 2a–d, f, Supplementary Fig. 2h–l) is small and seemingly has two lumina (Figs. 1b, 2a–d, f). This

**Fig. 1 | Jurassic lampreys from the Yanliao Biota, China, *Yanliaomyzon occisor* gen. et sp. nov. and *Yanliaomyzon ingensdentes* gen. et sp. nov.**
**a**–**e** *Yanliaomyzon occisor* gen. et sp. nov., **a** Photograph of holotype (IVPP V 15830); **b** Line drawing of the oral disc and dentition of (**a**), based on Supplementary Fig. 2k and l; **c**, **d** Paratype (IVPP V 18956B), photograph (**c**) and line drawing (**d**); **e** Restoration. **f**–**h** *Yanliaomyzon ingensdentes* gen. et sp. nov., **f** Photograph of holotype (IVPP V 16715B), white arrow pointing to the skeletal relics in gut content; **g** Oral disc and dentition; **h** Restoration. Abbreviations: adf, 'anterior dorsal fin' (dorsal fin); af, anal fin fold; ba, branchial apparatus; ca, cloaca (anus); cot,

circumoral teeth; da, dorsal aorta; dcf, dorsal lobe of caudal fin; dt, oral disc teeth; cf, caudal fin; e, eyes; dt, disc teeth; go, external gill openings; gp, gular pouch; ic, intestine contents; io, infraoral lamina; ll, longitudinal lingual lamina; ll.l, left longitudinal lingual lamina; ll.r, right longitudinal lingual lamina; lv, liver; ns, olfactory organ (nasal sac); oc, otic capsule; od, oral disc; of, oral fimbriae; op, oral papilla(e); paf, precloacal skin fold; pdf, 'posterior dorsal fin' (anterior part of caudal fin); pt, piston cartilage; so, supraoral lamina; tl, transverse lingual lamina; vcf, ventral lobe of caudal fin; V1?, ophthalmic ramus of trigeminal nerve?

structure was also observed in the Cretaceous lamprey *Mesomyzon mengae*[9] but never seen in any other fossil lampreys[3,8,21–27]. Where present, this structure in pre-spawning and spawning males may have the function of courtship display or energy storage for spawning migration for anadromous species[28]. Among living lampreys, seven out of the 12 species with gular pouch are anadromous[12,20,28].

The dorsal fin (adf) of *Yanliaomyzon* is fairly long and extends anteriorly until the level of the fourth gill pouch (Fig. 1c–e, f, h) as in *Mesomyzon mengae*[9]. Just since the position of the cloaca (ca), a long precloacal skin fold (paf) is developed in both species and extends anteriorly to the anterior branchial region (Fig. 1c–e, f, h).

In the position of the intestinal tract, several tooth-bearing jaw bones and possibly skull bones of some unidentified bony fishes and some skeletal relics (ic) are preserved in *Y. occisor* (Supplementary Fig. 2a, c–f) and *Y. ingensdentes* (Supplementary Fig. 1a, f), respectively. Together with the anatomy of the feeding apparatus described above, the bones and skeletal relics point to a flesh-eating habit for these fossil lampreys, making them the oldest records of its group with feeding mode clearly specified so far.

## Phylogenetic analyses
Calibrated by these Jurassic stem lampreys and based on an updated character data matrix (Supplementary Note 2), the Petromyzontiform timetree is revised and contrasts with those in other relevant studies[3,16,29–34], especially in the age and phylogeny of the crown-group lampreys. The all-compatible consensus tree shows that within the cyclostome monophyly and crownward to the basal Paleozoic lampreys, three Mesozoic species, *Mesomyzon mengae*, *Yanliaomyzon ingensdentes* and *Y. occisor* are resolved successively along the stem, with the latter two immediately basal to crown-group lampreys (Fig. 3). This is different from a recent study where *Mesomyzon mengae* is placed in the crown-group lampreys as the sister to Petromyzonidae[34]. In current crown-group interrelationships, *Geotria* (Geotriidae) was resolved as the earliest diverging lineage, sister to the pair of *Mordacia* (Mordaciidae) and Petromyzontidae, whereas other studies either support Geotriidae-Mordaciidae[29,34] or Geotriidae-Petromyzontidae pairs[16,32,33], or their interrelationships are unresolved in a basal polytomy[3,31]. *Mesomyzon*, the earliest lamprey with a modern triphasic life cycle known so far[3,9,13], diverged from other lampreys in 181.8 (236.0–163.00) Ma (median age and 95% HPD interval, same for below). The petromyzontiform crown was estimated to originate in 78.0 (122.7–40.8) Ma, which is roughly 90 million years (Myr) younger than the recent total-evidence inference (169.82 Ma)[29] and roughly 240–97 Myr younger than previous molecular-dating estimate (280–220 Ma)[30], respectively. Our estimation of the crown age is also ca. 140-95 Myr younger than those (171.59–219.29 Ma) in a recently reconstructed lamprey phylogeny[34]. Within the crown, the split of Southern and Northern Hemisphere lampreys was estimated to occur in 58.4 (95.0–29.5) Ma, which is again much younger than the aforementioned estimations[29,30,34]. The inferred phylogeny of Petromyzontidae is similar to the one presented by Brownstein and Near[34] but the age (ca. 26 Ma) of the most recent common ancestor (MRCA) of this family is again much younger than 94.1 (139.56–62.41) Ma in that study[34], with the appearance of most living Northern Hemisphere lamprey lineages

dated to after 10 Ma, still younger than the estimation (ca. 20 Ma) of the same study[34].

## The evolutionary rates of lampreys' feeding apparatus
Regarding the evolutionary rates of the feeding apparatus, the Mesozoic species act as the 'watershed' of the history of lampreys (Supplementary Discussion). The relative rates along the branches downward and crownward to themselves greatly differ, with the former almost ten times the latter (Supplementary Fig. 3). The evolutionary rate is rather low along the stem above *Yanliaomyzon* in the Late Jurassic, and then the rates are very uneven after the emergence of the MRCA of the crown-group lampreys, with the rate at the branch subtending Mordaciidae and Petromyzontidae ca. 77 times that low rate toward Geotriidae (Fig. 3). This evolutionary rate heterogeneity reflects the novelties of the feeding apparatus in the Mesozoic lampreys relative to the Paleozoic species and highlights the marked reinforcement of those *Geotria*-like biting structures in *Yanliaomyzon* species, which approximate the state of the MRCA of the crown-group lampreys. It also signifies the rapid diversification of the feeding habits during the early history of the living lamprey lineages (Fig. 3).

## The reconstruction of the ancestral feeding mode of living lampreys
Based on the characteristic reconstruction of the feeding mechanism (Supplementary Codes 4–6), the MRCA of all living lampreys probably had a feeding apparatus comprising spatulate oral disc teeth, a large supraoral lamina with two central cusps flanked by lateral flanges, a U-shaped movable transverse lingual lamina with three large cusps, and a parenthesis-shaped longitudinal lingual lamina. In general, the assembly of such a feeding apparatus is fairly similar to those in the feeding adults of pouched lamprey *Geotria australis*[10,11] and *Yanliaomyzon* species described here and points to a flesh-feeding habit[10,11] (Fig. 3).

## The ancestral distribution reconstruction
Reconstructions of ancestral areas were based on the assignments of fossil and extant lineages into nine geographic provinces (Fig. 4) (Methods and Supplementary Codes 7, 8) by incorporating the scheme proposed in a recent study[34]. The MRCAs of the Cretaceous (*Mesomyzon mengae*) and Jurassic (*Yanliaomyzon* species) plus the crown-group lampreys were all estimated to originate most likely in Eastern Laurasian landmass, with the MRCA shared by *Mesomyzon* having an Early Jurassic age and that shared by *Yanliaomyzon* a late Middle to early Late Jurassic age (Fig. 4). The MRCA of all living lampreys probably originated in the Southern Hemisphere (southern Australia plus New Zealand + South America, with a combined probability of ca. 40%) of the Late Cretaceous (ca. 78. Ma, middle Campanian). After the divergence of *Geotria*, the MRCA of *Mordacia* + Petromyzontidae (Northern Hemisphere lineages) was most likely (with a probability of ca. 40 %) also a Southern Hemisphere species with an age of early Cenozoic (ca. 58 Ma). The MRCA of the Northern Hemisphere lampreys (Petromyzontidae) was estimated to arise in Western North America (Northeastern Pacific) of the late Oligocene (ca. 26 Ma), followed by the divergence of two main lineages that began to diversify during the middle Miocene in Western North America (ca. 15 Ma) and

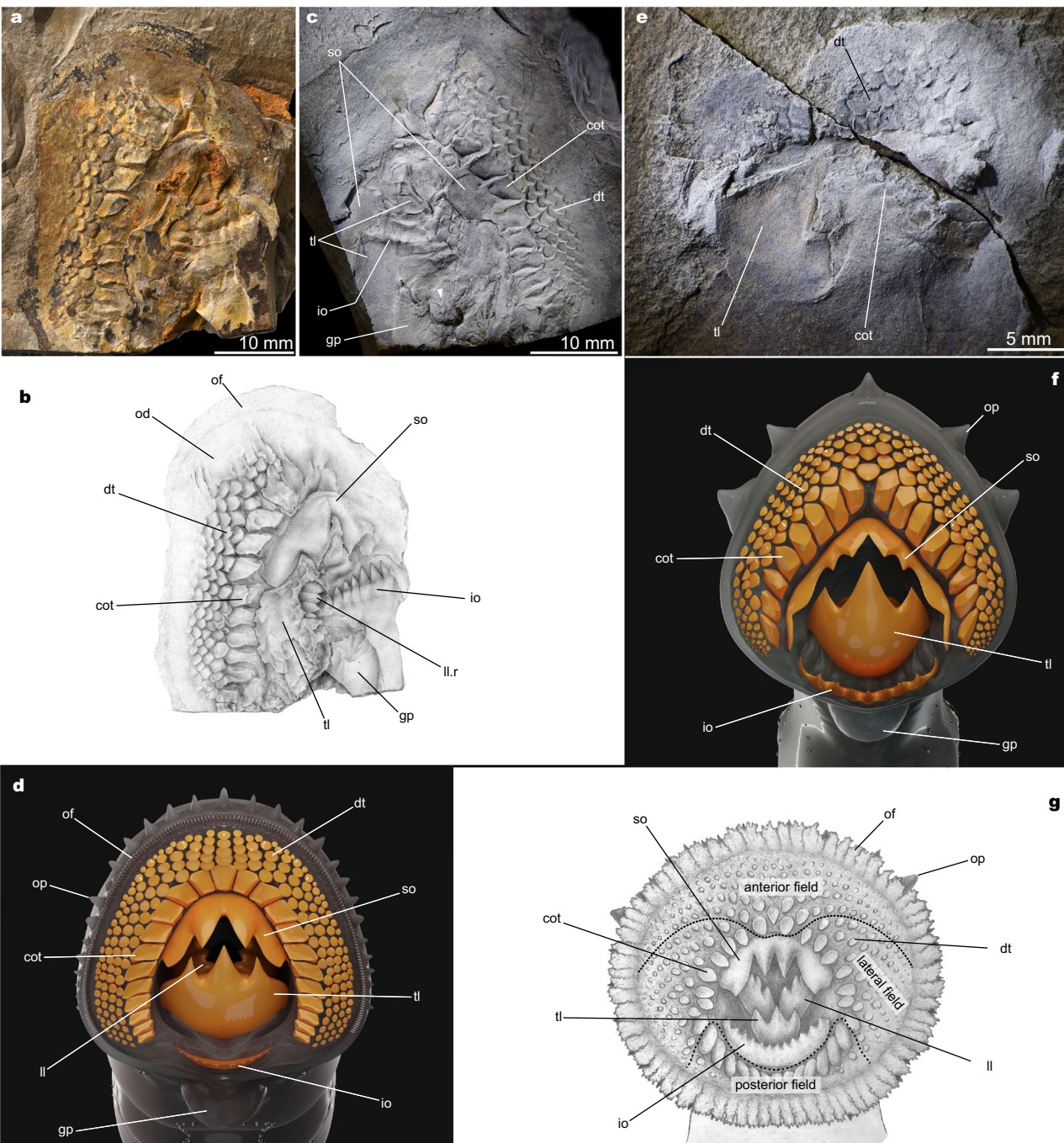

**Fig. 2 | Feeding apparatus of *Yanliaomyzon* gen. nov. and pouched lamprey *Geotria australis*. a–d** Oral disc and dentition of *Yanliaomyzon ingensdentes* gen. et sp. nov., **a** Photograph (IVPP V 16716B) and **b** Line drawing; **c** Photograph (IVPP V 16716A), whitened with ammonium chloride, the white arrow pointing to the imprints of the wrinkles of the gular pouch; **d** Restoration. **e, f** Oral disc and dentition of *Yanliaomyzon occisor* gen. et sp. nov., **e** Photograph (IVPP V18956A), whitened with ammonium chloride; **f** Restoration; **g** Oral disc and dentition of *Geotria australis*, redrawn from ref. 10. Abbreviations: cot, circumoral teeth; dt, oral disc teeth; gp, gular pouch; ic, intestine contents; io, infraoral lamina; ll, longitudinal lingual lamina; ll.r, right longitudinal lingual lamina; od, oral disc; of, oral fimbriae; op, oral papilla(e); so, supraoral lamina; tl, transverse lingual lamina.

in Eastern North America or Europe (Northern Atlantic) (ca. 13 Ma), respectively.

## Discussion

### Body size, life history and swimming performance of *Yanliaomyzon*

The body size of the Jurassic lampreys provides insights into the reconstruction of their life-history pattern. *Yanliaomyzon occisor*, to

our knowledge, the largest fossil lamprey known so far, ranks among the largest in modern species[7,12]. Living lampreys' adult body size is intrinsically related to their key biological features, with the larger and predaceous/parasitic species capable of migrating farther and achieving wider distribution, laying much more eggs, and being more tolerant to salt waters[35]. In body length, *Yanliaomyzon occisor* falls within modern anadromous lampreys and notably departs from the smaller freshwater residents and those species that do not feed post-

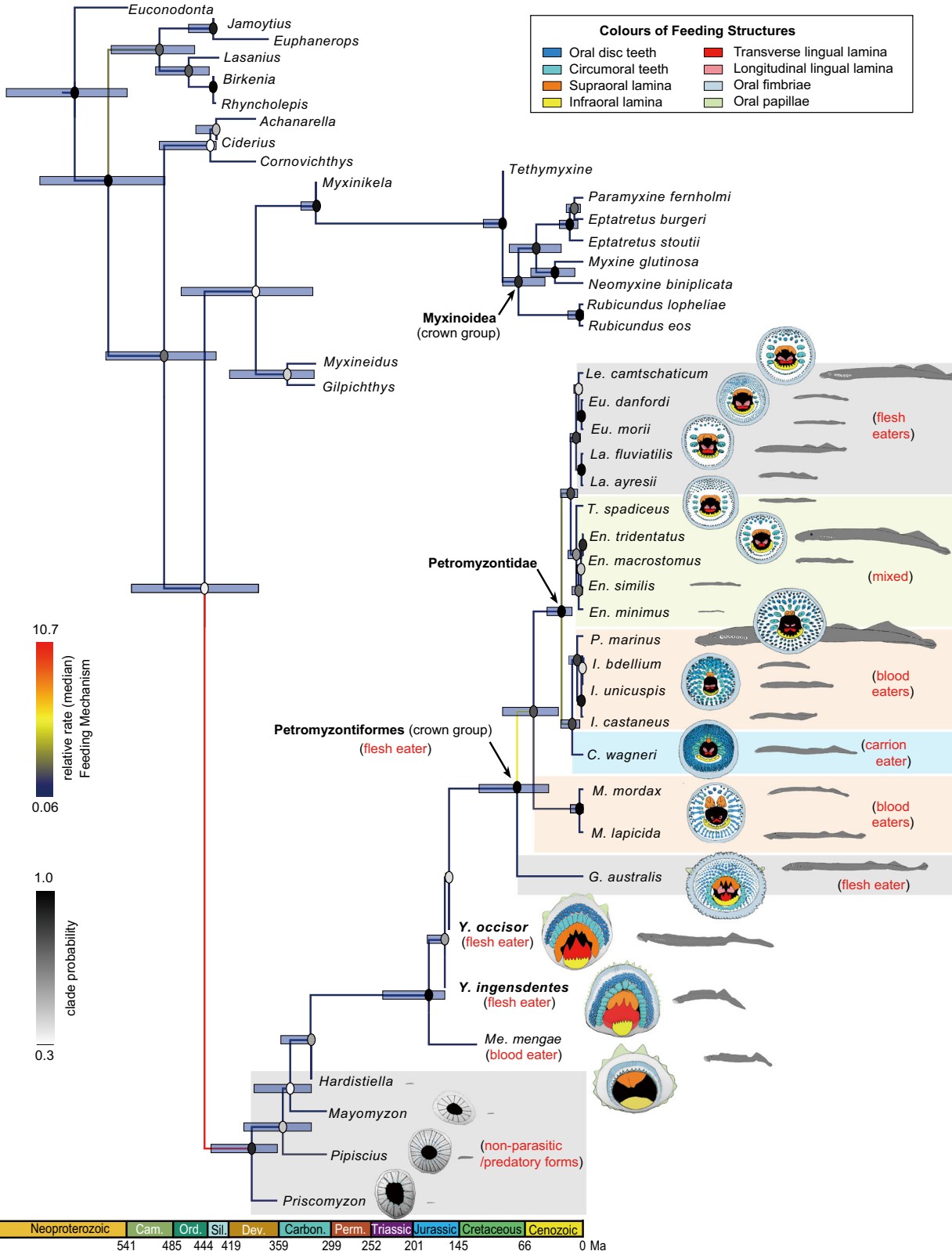

**Fig. 3 | Time-calibrated phylogeny of the cyclostomes and lampreys.** The timetree is the all-compatible consensus tree summarized from the Bayesian total-evidence dating analysis on the partitioned data (Supplementary Codes 1-3). The node ages in the tree are the posterior medians, and the error bars at the nodes denote the 95% highest posterior density (HPD) intervals. The shade of each circle represents the posterior probability of the corresponding clade. The color of the branch represents the median relative evolutionary rate of the feeding mechanism characters at that branch. The median values of the relative rates in some focal branches are shown in Supplementary Fig. 3. The oral disc and dentition were redrawn according to relevant literatures[3, 9–12, 39]. Abbreviations: *C., Caspiomyzon*; Cam., Cambrian; Carbon., Carboniferous; Dev., Devonian; *En., Entosphenus; Eu., Eudontomyzon; G., Geotria; I., Ichthyomyzon; La., Lampetra; Le., Lethenteron; M., Mordacia*; Ord., Ordovician; *P., Petromyzon*; Perm., Permian; Sil., Silurian; *T., Tetrapleurodon; Y., Yanliaomyzon*.

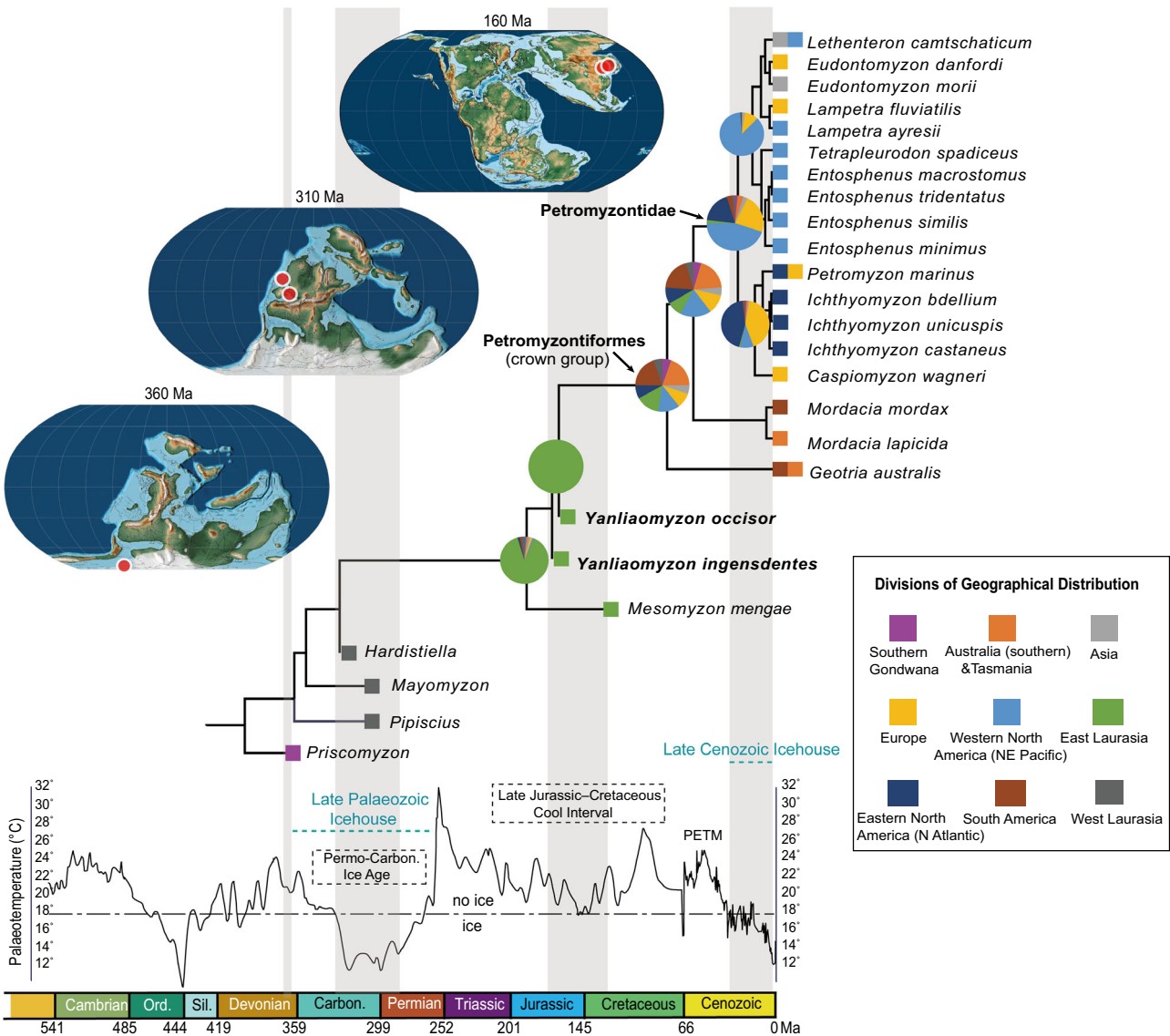

**Fig. 4 | Timetree of the Petromyzontiformes projected with paleotemperature curve since the Devonian and biogeographic reconstructions of the group.** Paleotemperature curve follows ref. 51. Paleogeographical maps are modified from ref. 48, on which fossil lamprey localities are marked as red spots. Source data are provided as a Source Data file.

metamorphosis[14], which implies that this large lamprey was probably anadromous. Therefore, the current phylogenetic distribution of anadromy in this stem lineage and the earliest diverged lineage of the crown group (*Geotria*) suggests that the MRCA of living lampreys was probably also anadromous. Being phylogenetically bracketed by *Mesomyzon* and modern lampreys, which all have a triphasic life cycle, *Yanliaomyzon* species could have likely inherited such a life-history strategy from their common ancestor shared by *Mesomyzon*, given that the acquisition of this trait is probably irreversible in lampreys[36].

*Yanliaomyzon* species, like *Mesomyzon*, also have an extraordinarily long-based dorsal fin and a ribbon-like precloacal skin fold that considerably extends the lateral side of the body and thus facilitates power transmission to the water during swimming[37]. As shown in living eels (*Anguilla vulgaris*) and African knifefish (*Gymnarchus niloticus*) with similar body plans, these vertical stabilizer structures represent an adaptation of swimming with high hydromechanical efficiency[37,38], which is especially significant in the upstream migration and long-distance dispersal for lampreys.

## The evolution of lampreys' feeding apparatus and its implications

As indicated by the lineage of *Mesomyzon* in our Petromyzontiform timetree, the substantial shift of lampreys' life cycle pattern from the Paleozoic type without ammocoete stage[3,15] to that comprising three stages might have occurred no later than the late Early Jurassic (ca. 180 Ma) (Fig. 3). This evolutionary breakthrough coincided with a dramatic evolutionary increase of the body size of these animals (Fig. 3), a natural consequence of the evolutionary dynamics of host-lamprey interactions and the changing ecological structures of the ichthyofaunal communities.

Lampreys underwent radical changes in their feeding structures and feeding habits since their appearance in the Devonian (Fig. 3). This history can be divided into two episodes linked by the Jurassic species (Supplementary Discussion). The Paleozoic lampreys might not hold parasitic or predaceous habits as conventionally assumed[3,15]. They are all very small and lack an ammocoete stage[3,15], which should be associated with their simply structured and tiny dentition and the limited buccal cavity, the space accommodating the anti-coagulant secreting

glands and food processing in living lampreys[6]. Their buccal cavity is squeezed by the anteriorly positioned branchial apparatus, which culminates in *Priscomyzon*[3,26], whose first two gill pouches even extend anterior to the eye[3]. These early lampreys' oral disc (e.g., the extra-ordinarily large oral disc of *Priscomyzon*[3]) and associated structures (e.g., the radiative arrangement of the circumoral petaliform plates of *Priscomyzon*[3], *Pipiscius*[3], and probably *Mayomyzon*[39] that would add the rigidity of the disc) all point to the enhanced attaching capability, but the real 'biting' structures, i.e., those laminae on the gouging pis-ton, or the supraoral lamina which might function in feeding were so weakly developed that they were never preserved or unrecognizable in the fossils despite the existence of the clearly observable circumoral teeth in situ[3]. Moreover, were they parasitic or predatory forms, their feeding opportunities were rather limited because the vast majority of their potential hosts then all had thick scales or armor surfaced by enamel or dentine layers or ridges[40,41], thus too hard for lampreys' keratinous teeth to penetrate. Instead, we propose that these early lampreys' well-developed oral discs and attaching skills might be adapted to scraping algal mats on other aquatic animals, thereby opening up a new niche to avoid the competition from the con-temporary diversified jawless fishes[40], especially that from the long-bodied jawless conodonts with comparable feeding apparatus[42]. Also, their oral discs could be used for hitch-hiking on other aquatic verte-brates to expand their distribution, which could partially account for the wide historical distribution of these animals from the polar region of the Late Devonian to the low-latitude region of the Late Carbon-iferous (Fig. 4). Lampreys' oral disc could also be involved in some reproductive behaviors, e.g., moving stones for nesting or anchoring to the stone or the partner female during the mating as in living lampreys[5].

The post-Paleozoic lampreys made a breakthrough in their evo-lutionary history in the Early Jurassic, underlain by the structurally complicated feeding apparatus and the emergence of new food resources. Accumulated by the evolution since the Carboniferous, the feeding apparatus of the ancestor giving rise to the lineage of *Meso-myzon* in the Early Jurassic had been enhanced by developing more complicated oral disc appendages and large supra- and infraoral laminae (teeth) in addition to the circumoral teeth[9] (Fig. 3). Meanwhile, they encountered a critical evolutionary opportunity offered by their most significant potential hosts, i.e., the derived teleost fishes includ-ing leptolepid fishes plus more advanced teleost fishes that abundantly emerged with the thinner cycloid scales replacing the once-dominant ganoid scales and prevailing since the late Early Jurassic[43,44], coupled with the appearance of early members of current lampreys' host lineages, e.g., the acipenseriform fishes[45]. Besides, the vanishment of their potential competitors might give way to lampreys' evolution. The jawless conodonts, which have similar general morphology and feed-ing anatomy[42,46], went extinct in the latest Triassic and probably released the ecological niche subsequently occupied by the lampreys in the Jurassic period. Therefore, provided with better-armed feeding structures and increased feeding opportunities, as well as possibly new ecological space, lampreys from the Jurassic onward had evolved sufficiently large body size. We herein hypothesize that this novelty, together with the enhanced swimming capability represented by Jur-assic and Cretaceous species, should have rendered the lampreys elevated hydromechanical advantage and reproductive output to withstand the distance, rigor and mortality of the upstream migration. In effect, these paved the way for lampreys' habitat expansion to the freshwater system and fueled the innovation of the life-history strategy by introducing a radical metamorphic stage.

The reinforcement of the feeding apparatus peaked in the Middle and Late Jurassic lampreys, which prefigured the ancestry of the feeding mode of living lampreys (Fig. 3). *Yanliaomyzon*'s dentitional arrangement and the gut content point to a flesh-eating habit. Their dentitional pattern resembles *Geotria*, a large flesh eater that can even

destroy the skull of teleost fish[46]. Our study suggests that the flesh-eating habit is likely ancestral for modern lampreys, rather than the blood-eating as traditionally noted[5,11,14,46], nor the non-feeding habit[33]. The establishment of this feeding habit in Jurassic lampreys imple-mented their energy intake and growth potential as in their living counterparts[7], which should have fueled their long-distance dispersals and exerted a subtle influence on the later history of the group, including the origination and distribution of modern lineages.

## The historic biogeography of modern lampreys

Living lampreys' anti-tropical distribution is effectively restricted in temperate or sub-arctic regions, north and south of the 30° parallel and marked out by the annual 20 °C isotherm[5]. The adaptation to cool or cold waters might also exist in early lampreys. The paleo-temperature curve projected to our dated petromyzontiform timetree (Fig. 4) shows that the emergence of recorded fossil lampreys[9,21–27] appears to coincide with the cool or cold geological intervals and came from productive areas influenced by continental ice sheets, cooling events or topographical highland[17,47–52] (Supplementary Discussion). This could suggest that lampreys' environmental preference might have remained static throughout their history and hence being more frequently discovered in cool settings.

The ancestral area reconstructions outlined the scenarios of the historic biogeography of the post-Paleozoic lampreys (Fig. 4). Trans-equatorial dispersals from the Northern to Southern Hemisphere after the age of *Yanliaomyzon* were inferred before the emergence of the crown-group lampreys (Fig. 4). Given early lampreys' occurrences in the Euramerican paleoequatorial regions in the Late Carboniferous Ice Age[49,50], such a scenario was fairly possible during the cool intervals in the Tithonian-early Barremian Cool Interval and the Aptian-Albian Cold Snap[51] when some cold-adapted nannofossil species had dispersed from high-latitude to low-latitude regions[53]. As for lampreys, the increased body size (e.g., *Yanliaomyzon occisor*) at this evolutionary level must have given them long-distance dispersal capability just as the cases in the large living species[5,14] and therefore made this voyage more manageable, just like the case of pouched lamprey *Geotria aus-tralis* and Pacific lamprey *Entosphenus tridentatus* now inhabiting both sides of the southern and northern Pacific Ocean, respectively[12,16,20,33]. It would be not surprising if *Yanliaomyzon* and *Mesomyzon* had wider distributions because they all had a much longer dorsal fin and hence higher hydromechanical efficiency than the pouched lamprey and Pacific lamprey. As extant predatory lampreys, e.g., the pouched lamprey and the Pacific lamprey, may follow the movements of fish shoals over considerable distances[5] and in varying water depth (com-monly 0–500 m, maximum record 1485 m)[54], in this 'Isothermal Sub-mergence' pattern of dispersal hypotheses for anti-tropicality biogeography[55], the diversification and global distribution of the potential host fishes from the Late Jurassic to Late Cretaceous periods[44] could have also led lampreys' trans-equatorial dispersals.

The revised interrelationships of crown-group lamprey lineages shed light on the formation of modern lampreys' distribution pattern. The southern lampreys were conventionally proposed as the deriva-tives of Northern Hemisphere lineages via a 'pre-Tertiary' departure[1,5,47]. With the advent of cladistics and molecular clock, the petromyzontiform crown age was estimated to range from 280–220 Ma[30] to 170 Ma[29], hence predating the tectonic splits of the Gondwana[48]. Living lampreys' 'poles-apart' distributional pattern (or 'anti-tropicality'[55]) was also attributed to the drift vicariance related to the breakup of Pangea ca. 200 Ma into northern and southern landmasses[5,34]. Contrary to past efforts, our study points to the Southern Hemisphere as the biogeographic source for modern lam-preys and pushes the ages of the petromyzontiform crown and southern-northern divergence to much younger intervals. These ages correspond to two relatively cool intervals, with the former marking the ending of the Cenomanian-Turonian Thermal Maximum, which

might have wiped out the stem lamprey lineages, and the latter culminating immediately in the Paleocene-Eocene-Thermal-Maximum (PETM)[51]. The warmed tropical and low-latitude seas and the declined equator-to-pole temperature gradient during PETM[56] and later warm periods[51,57] should have greatly restricted the distribution of these cold-water animals and ultimately shaped their anti-tropical, disjunct biogeographical pattern.

The Northern Hemisphere lampreys diversified in concert with the rhythm of the contemporaneous climatic fluctuations and the ichthyofaunal changes during the Cenozoic. The MRCA of Northern Hemisphere lampreys (Petromyzontidae) was inferred to arise from the trans-equatorial dispersal of some stem petromyzontid lampreys from the Southern Hemisphere between the latest Paleocene to the late Oligocene, an interval including a global cool phase in the early Oligocene[57]. The Petromyzontidae was estimated to have their origin in the late Oligocene of Western North America and began to diversify after the earth entered an 'ice-house' phase from the 'greenhouse' phase[58] (Fig. 4) and the Greenland glacial ice and pan-Arctic sea ice first emerged[59]. Notably, this timing is also compatible with the diversification of the major fish and aquatic mammalian hosts of modern lampreys[58,60–63]. After the Mid-Miocene Climatic Optimum (17–15 Ma), the diversifications of living lamprey accelerated and occurred both in Western North America (Northeastern Pacific) and Eastern North America or Europe (North Atlantic), especially when they evolved under a gradual cooling phase initiated at ca. 10 Ma[57] (Fig. 4). The favorable climatic settings and increased food supply boosted the diversification of lampreys in the Northern Hemisphere and eventually the establishment of uneven species richness between two hemispheres, which is accommodated by more numerous and diversified riverine systems in northern temperate regions than in corresponding areas of the Southern Hemisphere[16].

Together, fossil lampreys herein suggest that its group is not as conservative as previously thought, and the innovations of their feeding biology had probably underlain their evolutionary increase of the body size and the 'modernization' of their life-history mode during the Jurassic period. Extant lampreys are geologically fairly young with their basic biogeographical pattern, i.e., the anti-tropical distribution, sourced from the Southern rather than Northern Hemisphere and proving to be historic products of the biological and climatic changes since the Late Cretaceous.

# Methods

## Fossil specimens
This research complies with all relevant ethical regulations. This study is based on two specimens of *Yanliaomyzon occisor* gen. et sp. nov. and two specimens of *Yanliaomyzon ingensdentes* gen. et sp. nov., which are housed at the Institute of Vertebrate Paleontology and Paleoanthropology (IVPP), Chinese Academy of Sciences. The specimens of *Yanliaomyzon occisor* gen. et sp. nov., IVPP V 15830 comes from the Tiaojishan Formation, earliest Late Jurassic, ca. 158.58–160 Ma[19] of Daxishan, Toudaoyingzi Town Jianchang County, Liaoning Province, and IVPP V 18956A, B from the corresponding layers of Nanshimen Village, Gangou Town, Qinglong County, Hebei Province, China. The specimens (IVPP V16715A, B, IVPP V 16716A, B) of *Yanliaomyzon ingensdentes* gen. et sp. nov. come from the Daohugou beds, late Middle Jurassic, ca. 163 Ma[19] in Wubaiding Village, Reshuitang County, Liaoning Province, China. The materials were collected by Xiaoling Wang and Min Wang of the Institute of Vertebrate Paleontology and Paleoanthropology (IVPP), Chinese Academy of Sciences.

## Data
Different from the recent comprehensive vertebrate and cyclostome phylogeny[29], this study focuses on resolving the systematic positions of the Jurassic fossil lampreys within the petromyzontiform as well as their closely related taxa, thus the remote taxa were not included, e.g.,

the hemichordate (*Saccoglossus kowalevskii*), tunicate (*Ciona intestinalis*), cephalochordate (*Branchiostoma floridae*), *Pikaia*, controversial Cambrian vertebrates (*Haikouella*, *Haikouichthys*, *Metaspriggina*, and *Myllokunmingia*), and the lineage leading to the gnathostomes (jawed vertebrates). In this context, the dataset includes 25 extant taxa and 20 fossils from Cyclostomi (total group) and combines both morphological characters and molecular sequences, with morphological characters for all taxa (208 characters, see Supplementary Note 2), 16S gene for 10 extant species (772 sites) and CO1 gene for 11 extant species (721 sites). The character description and revision are listed in Supplementary Note 2.

The Bayesian total-evidence dating analysis was performed using the software MrBayes 3.2.8[64]. This version was not formally released when performing this study and the executable was compiled from the latest source code (https://github.com/NBISweden/MrBayes). We describe the models, priors and Markov chain Monte Carlo (MCMC) settings for the analysis. The data matrix and MrBayes commands are given in Supplementary Codes 1 and 2.

## Substitution models
The Mkv model[65] was used for the morphological data correcting for the variable coding bias (only variable characters are coded in the matrix). For each gene, the HKY model[66] was used with an independent transition-transversion rate ratio ($\kappa$) and exchangeability rates. Rate variation across characters (sites) was modeled using the discrete gamma model[67], while rate variation across data partitions was drawn from a uniform Dirichlet distribution[68] under the constraint of the average rate across characters (sites) being 1.0.

We used the default priors for the substitution parameters, e.g., Exponential(1.0) for the gamma shape parameter ($\alpha$), Beta(1, 1) for the transformed transition-transversion rate ratio ($\kappa/(1+\kappa)$), and a uniform Dirichlet distribution for the exchangeability rates ($\pi$).

## Total-evidence dating
Traditional phylogenetic analyses infer branch lengths measured by expected changes per character, which are products of geological times (myr: million years) and evolutionary rates (expected changes per character per myr). Without the fossil ages and clock-model assumptions, the two components are confounded and indistinguishable. In the total-evidence dating analysis, we use fossil ages as tip calibrations to co-estimate divergence times and evolutionary rates. To investigate evolutionary rate heterogeneity among separate morphological regions and between morphological characters and molecular sequences, the morphological characters were divided into three partitions, that is, feeding mechanism (51 characters), branchial apparatus (22 characters) and others (135 characters) (Supplementary Code 2). Eventually, each of the three morphological partitions had an independent clock model, and the two gene partitions share another independent clock model.

All partitions share the base clock rate ($c$), which was assigned a lognormal (−6, 1.0) prior with mean 0.004 and standard deviation 0.005. The evolutionary rate variation was modeled by the relative rate ($r$) multiplied by the base rate ($c$). The relative rates in each of the four relaxed clocks follow independent and identically distributed (i.i.d.) lognormal distributions[69] with mean 1.0, and the variance of each clock ($\sigma$) was assigned an Exponential(1.0) prior by default.

Apart from the clock model, the other component is the timetree. The timetree was modeled by the fossilized birth-death process[68,70,71] with speciation rate $\lambda$, extinction rate $\mu$, fossil-sampling rate $\psi$, extant-sampling proportion $\rho$, and conditioned on the root age ($t_{mrca}$). The ages of the fossils were fixed to their geological times (see Supplementary Code 2). For the priors, $t_{mrca}$ was assigned an offset-exponential distribution with mean 600 and minimum 500 Ma referring to the oldest fossils, $d = \lambda - \mu$ ~ Exponential(100) with mean 0.01, $v = \mu / \lambda$ ~ Uniform(0, 1) and $s = \psi / (\mu + \psi)$ ~ Uniform(0, 1). The extant-

sampling proportion was fixed to 0.3, assuming 25 out of about 80 living species were sampled and included in the analyses.

Two independent MCMC runs with 4 chains per run (1 cold, 3 heated) were run 30 million generations and sampled every 400 generations. The first 25% samples were discarded as burn-in and the remaining were combined after examining the convergence of both runs. The posterior tree samples were summarized as an all-compatible consensus tree (Supplementary Code 3), which displays all compatible clades in the posterior trees as bifurcations, including those with a probability smaller than 0.5.

## Ancestral state reconstruction

To study the evolution of lampreys' feeding modes, we inferred the ancestral states of the feeding mechanism characters for the node of the most recent common ancestor (MRCA) of the living lampreys (crown group). This inference was done in the same total-evidence dating framework described in the previous section, except for additionally constraining the crown group as a monophyletic clade (Supplementary Codes 4–6).

We also reconstructed the ancestral geographical ranges for several key nodes, including that of the MRCA of crown lampreys (Fig. 4). The geographical locations of the living lampreys were coded into nine provinces (0. South Africa (southern Gondwana); 1. Australia (southern) & Tasmania; 2. Asia; 3. Europe; 4. Western North America (Northeastern Pacific); 5. East Laurasia (North China Craton); 6. Eastern North America (Northern Atlantic); 7. South America; 8. West Laurasia) (Supplementary Codes 7, 8) in addition to the scheme proposed by a recent study on the phylogeny and historic biogeography of modern lampreys[34]. In our analysis, the geographical distributions of some living species were revised from the assignments in ref. 34: *Caspiomyzon wagneri* was coded as a European taxon rather than an Asian one[34] based on their geographical isolation from the distribution of the species in temperate Asia[4,12] and the European distribution of the congeneric species[34]; *Entosphenus tridentatus* was coded to be distributed in both Western North America and Asia because it is also distributed in Japan[12], so was *Lethenteron camtschaticum*[12]. We used the subtree of the stem and crown lampreys (rooted at *Priscomyzon*) inferred from the total-evidence dating analysis and fixed the node ages to their median estimates.

## CT scanning and computerized tomography

X-ray micro-computerized tomography was adopted for scanning the oral disc and the dentition of IVPP V 15830. The scanning was carried out using the 225 kV micro-computerized tomography (developed by the Institute of High Energy Physics, Chinese Academy of Sciences (CAS)) at the Key Laboratory of Vertebrate Evolution and Human Origins, CAS[72]. The specimen was scanned with a beam energy of 140 kV and a flux of 120 µA at a resolution of 36.07 µm per pixel using a 360° rotation with a step size of 0.5°. A total of 720 projections were reconstructed in a 2048*2048 matrix of 1536 slices using a two-dimensional reconstruction software developed by the Institute of High Energy Physics, CAS. The data then was analyzed by using VGstudio 2.2 to produce the 3D images (Supplementary Fig. 2l).

## Illustrations

All drawings of Figs. 1 and 2 were done with pencils by the author. To distinguish the types of dentition and the associated structures of the feeding mechanism, different colors were filled in corresponding regions using Adobe Photoshop CS6. The phylogenetic trees were visualized in FigTree (v1.4.4) and adjusted in Adobe Illustrator CS6 and Affinity Designer (v1.8.4). The digital reconstructions and models (Figs. 1e, h, 2d, f) were done by NICE Vistudio (Paleovislab, IVPP).

## Reporting summary

Further information on research design is available in the Nature Portfolio Reporting Summary linked to this article.

## Data availability

All fossils are curated at the fossil collections of the Institute of Vertebrate Paleontology and Paleoanthropology (IVPP), Chinese Academy of Science. The data underlying this article are available in its online Supplementary Information. Source data are provided with this paper.

## Code availability

The codes used for the study are provided in the Supplementary Code files. Statistical analyses were performed using the Bayesian phylogenetic inference software MrBayes (https://github.com/NBISweden/MrBayes).

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

## Acknowledgements
We thank the organizers of Second Tibetan Plateau Scientific Expedition (STEP) for financial and logistic support during the fieldwork and data collection. We thank M.-M. Chang for her enlightening discussions; Special thanks to X.L. Wang and M. Wang for specimen collection; Thanks to D.S. Miao for stylistic improvement of the abstract of the early draft. We appreciate the help of H.B. Wang and Z.Q. Yu during the fieldwork and Y.M. Hou, C. Zhang for producing computed tomography and W. Gao and Z. Feng for photographing; Y.F. Chen for fossil preparation. We thank Q.M. Qu, D.Y. Huang, Y.A. Zhu, T. Miyashita and D. Suzuki for helpful discussions, L.P. Dong, C.C. Liao, Z.Q.G. Jiang, S.F. Li, and H.J. Song for providing some literatures, H. Li for assisting in data collecting and G.H. Xu for helping with photographing. We thank H.M. Zhang of NICE Vistudio (Paleovislab, IVPP) for designing and restoring 3D models. This work was supported by National Natural Science Foundation of China (42288201, 42272013, 42172006); C. Zhang was supported by the Hundred Young Talents Program of Chinese Academy of Sciences (Y902061) and P. Janvier by Chinese Academy of Sciences President's International Fellowship for Distinguished Scientists (2019DB0022).

## Author contributions
F.X.W. designed this study, prepared figures and wrote and revised the manuscript; P.J. contributed to organizing and revising the manuscript; C.Z. performed analyses and contributed to drafting the manuscript, especially the "Methods" section.

## Competing interests
The authors declare no competing interests.
