## [Peer Review File · Nature Communications]

The rise of predation in Jurassic lampreysReviewers' Comments:

Reviewer #1:

Remarks to the Author:

This is an outstanding contribution to our knowledge of lamprey phylogeny and biogeography through the integration of all known fossil and virtually all extant lampreys (*Geotria macrostoma* excluded) that feed post-metamorphosis. The discovery of two new fossil lampreys has enabled this breakthrough.

I have a number of comments and suggested changes to improve the manuscript.

The authors refer to the Jurassic era, but shouldn't it be Jurassic Period?

In the Abstract, the three phases of the life cycle should be specified for clarification.

The word "hosts" at the end of the Abstract and Introduction should be replaced by prey because, given their oral and lingual dentitions, the two newly-described fossil lamprey species were more likely predators rather than parasites.

In the Abstract, the word "the" should precede Pangean supercontinent.

In the Introduction, it is unclear to me what is meant by "simply assembled analogue". This needs to be clarified.

In the Introduction, the first occurrence of the expression "feeding mechanism" should be replaced by feeding apparatus.

In the Introduction, replace "the modern-type life history mode" by "one of the modern-type life history modes". There are four modes of life in extant lampreys: blood feeders (parasites), flesh feeders (predators), blood and flesh feeders (intermediates), and a presumed carrion feeder (scavenger) (Renaud and Cochran 2019). The full reference is: Renaud C.B. and P.A. Cochran. 2019. Post-metamorphic feeding in lampreys. pp. 247-285. In: *Lampreys: Biology, Conservation, and Control*. vol. 2, M.F. Docker (ed.), Fish and Fisheries Series 38, Springer Nature B.V. https://doi.org/10.1007/978-94-024-1684-8_3

At the end of the Introduction, change "renewed" to "revised" lamprey crown-group phylogeny. Under Systematic Palaeontology, the oral disc size is not admissible as a diagnostic character for the new genus because only one of the two new species has a measurable oral disc (see Supplementary Table 1). Notwithstanding, the oral disc is stated to be relatively small (at least in *Y. occisor*); compared to what? The oral disc size should be quantified and compared. How do you know that the teeth are made of keratin? It is sufficient to say posterior disc teeth lacking (i.e., to include and "posterior" circumoral is implicit). I agree that the anterior and lateral circumoral teeth are spatulate in shape, but cannot understand what is meant by "and each bearing three convergent blades". Figs 2d and f appear to be mislabeled. According to the species' diagnoses, Fig. 2d would correspond to a drawing of *Y. ingensdentis* and Fig. 2f to a drawing *Y. occisor*. In the figure caption, however, both Figs 2d and f are stated to correspond to *Y. occisor*, respectively, as a photograph and a restoration. I disagree with the limits of the anterior and lateral fields delineated in Fig. 2g. According to Renaud (2011: 7) (reference 11 in the manuscript), the boundaries are set by the anteriormost edge of the supraoral lamina and a line drawn through the apices of the two lateralmost cusps on the infraoral lamina. These limits, as defined by Renaud (2011), have repeatedly been used in the literature for counts of lamprey teeth, including those of *Geotria*, which the authors state resemble those of *Yanliaomyzon*.

In the diagnosis for *Y. occisor*, the tail region is stated to measure about one fourth of the total body length, but according to Supplementary Table 1, this rather equals 38.7%! (i.e., $a-C/TL = 270.7/700 = 38.7\% > 25.0\%$).

In the etymology for *Y. ingensdentis*, I would change rasping piston to gouging piston, because the latter is the action of a flesh feeder, while the former is the action of a blood feeder.

In the diagnosis of *Y. ingensdentis*, the dorsal fin is said to be contiguous with the high and large anterior caudal fin, but I believe the interpretation of the fins is incorrect. Rather, the low first dorsal fin is contiguous with the high second dorsal fin and the latter is continuous with the caudal fin.

In the Description and comparison, it is stated that *Y. occisor* is > 700 mm, but Supplementary Table 1 indicates exactly 700 mm, and therefore, the symbol greater than (i.e., $>$) should be deleted. This new species is said to be only slightly smaller than *P. marinus* (1,200 mm) and *E. tridentatus* (850

mm), but in this group should also be included *G. australis* (788 mm) and *L. camtschaticum* (790 mm) (see Renaud and Cochran 2019). Given these data, the qualifier "slightly" is incorrect and should be removed. The relatively small oral discs of *Y. occisor* and *Y. ingensdentis* are stated to be respectively, 4.8% and 6.7% of total length, but the former is correctly 4.7% and I can't find the data to derive the latter because Supplementary Table 1 does not have a value for the disc length of *Y. ingensdentis*. To state that the oral disc length to total body length of the two *Yanliaomyzon* species approach the condition of modern flesh-eating lampreys is not informative without providing the data. The data in Potter and Hilliard (1987) give a range for this ratio of 2.7–6.3% for flesh feeders (and specifically, 2.7–3.2% in *Geotria*) and 5.8–11.8% for blood feeders. As mentioned above, to state that posterior disc teeth are missing is sufficient because posterior circumoral teeth are by definition disc teeth. As also mentioned above, the meaning of the longest of the three converging blades facing slightly sideways is unclear to me. The expression tricuspid cusps is nonsensical (a tooth or a lamina may be tricuspid, but not a cusp). It would make more sense to say The tips of these three cusps ...

The prebranchial region is stated to be longer than the branchial apparatus in the two fossil lampreys; and this is certainly borne out in *Y. ingensdentis* (Supplementary Table 1: $d-B1 = 42$ versus $B1-B7 = 12$), but neither of these morphometric characters is recorded in *Y. occisor*. The eyes are stated to be medium-sized, but while a value of 4 is recorded in Supplementary Table 1 for *Y. ingensdentis*, none is given for *Y. occisor*.

The gular pouch is supposed to present in Fig. 2c and in Extended Data Fig. 2k, but this is certainly not evident to me. It is present in Fig. 2e, but only indicated with the abbreviation g and is missing the associated letter p, as defined in the anatomical abbreviations. According to references 11 and 22, there are in fact 11 (not nine) living lamprey species with a gular pouch, and six (not five) of these are anadromous. However, you also need to add *Geotria macrostoma*; making 12 species with a gular pouch, seven of which are anadromous. The reference for *G. macrostoma* is: Baker, C.F., C. Riva Rossi, P. Quiroga, E. White, P. Williams, J. Kitson, C.M. Bice, C.B. Renaud, I. Potter, F.J. Neira, and C. Baigún. 2021. Morphometric and physical characteristics distinguishing adult Patagonian lamprey, *Geotria macrostoma* from the pouched lamprey, *Geotria australis*. PLoS ONE 16(5): e0250601. <https://doi.org/10.1371/journal.pone.0250601>

Instead of Before the cloaca, a long preanal skin fold ..., I suggest A long precloacal skin fold ...
I suggest intestinal tract rather than intestinal duct.

Under Feeding mode, life history and swimming performance of *Yanliaomyzon* change stomach to intestinal or gut content.

The sentence This implies for the anadromy wherein the higher marine productivity ... must be missing some words because it makes no sense.

As suggested above, the preanal skin fold should be called precloacal skin fold. The long-based dorsal fin and ribbon-like precloacal skin fold considerably extend rather than extends.

Under The evolution of lampreys' feeding structures and its implications, I do not understand what is meant by hitch-hiking on other delivers. I suspect the word delivers is not what you mean.

I would term the radical metamorphic phase, a stage rather than a phase.

Under The early biogeographical history of modern lampreys, shouldn't Late Jurassic and Late Cretaceous be periods instead of eras?

I believe that the word promoted in relation to trans-equatorial dispersal is inappropriate and should maybe be replaced by led to.

I question the relevance of reference no. 48 in relation to major fish hosts of modern lampreys because in their revision of post-metamorphic feeding in lampreys, Renaud and Cochran (2019) do not mention the genus *Trisopterus* as a host of lampreys.

Under Anatomical abbreviations, the adf, anterior dorsal fin should be called first dorsal fin and the pdf, posterior dorsal fin should be called second dorsal fin. How can the posterior dorsal fin be synonymous with the anterior part of the caudal fin? I note that in lines 237-240 of the Supplementary Information, the authors have justified their position in regards to dorsal and caudal fin definitions, but to me it remains unsatisfactory. While in *Petromyzontidae* the second dorsal fin (all genera minus *Ichthyomyzon*) or posterior lobe of the single dorsal fin (*Ichthyomyzon*) is confluent with the caudal fin, *Geotria australis* is known to have a distinct second or posterior dorsal fin, separated by an interspace from the caudal fin.

Note that in reference no. 28, the second author is Gill, not Gills.

Note that in reference no. 34, the word Environments should have its first letter in lower case.

Note that in reference no. 42, the word Pacific should have its first letter in upper case.

Figs 3 and 4 contain much valuable information, but are too small for readers to be able to discern many of the details and should be increased in size. In regards to Fig. 4, it is unclear to me to which fossil taxa the four red spots refer to. Additionally, the distinction between the light blue and dark blue shadows is indiscernible.

The title of Supplementary Table 1 is misleading and confusing. A meristic measurement is a misnomer. The only meristic character it contains is the number of trunk myomeres and this has only been determined in *Mesomyzon menseae*. I suggest deleting Meristic from the title and simply starting with Measurements and removing the column number of trunk myomeres. I presume that all measurements are in mm, but this should be specified.

For completeness and balance of opinion, I strongly recommend the inclusion of two recent papers into the Discussion; namely, Brownstein and Near (2023) and Mallatt (2022). The complete references are: Brownstein, C.D. and T.J. Near. 2023. Phylogenetics and the Cenozoic radiation of lampreys. *Current Biology* 33: 1-8; Mallatt, J. 2022. Vertebrate origins are informed by larval lampreys (ammocoetes): a response to Miyashita et al., 2021. *Zoological Journal of the Linnean Society* XX: 1-35.

Claude B. Renaud

Reviewer #2:

Remarks to the Author:

This paper presents a radical revision of the evolutionary and ecological history of lampreys based on an exceptionally preserved fossil from the Jurassic. Prior fossils had suggested that the feeding morphology of lamprey was highly constrained for the last 360 million years with the exception of the pouched lamprey, and that key features like ammocete larvae evolved at some unknown time. The present study uses new data from a Jurassic fossil to support an abrupt change in lamprey ecology around the Mesozoic origin of the crown, with flesh-eating as the ancestral state and greater diversity in feeding morphology than anticipated. The study is very well written and wide ranging and thorough in its methods, implications for the evolution of fishes, and in the number of factors considered as drivers for the diversification of lamprey.

However, the one, somewhat glaring, omission is a lack of consideration of conodonts, which recently have been assigned to the cyclostome stem in other studies of fossil cyclostomes. Conodonts were flesh-eaters with feeding apparatus that sometimes resemble the novel plates of the new lamprey. They also finally went extinct at the end-Triassic. It seems possible that the extinction of sometimes large-bodied (>1m), pelagic, shearing-biting conodonts may have opened space for the previously non-biting lamprey (although conodonts were never freshwater). Conodonts are often ignored in evolutionary studies of fishes, but in this case there could be a real influence on the timing of lamprey diversification and the change in their ecology proposed here. I would like to see some discussion of this possibility.

In any case, this discovery and resultant investigation raises a number of interesting possibilities for and questions about Mesozoic fish evolution, which will lead to further reconsideration of the record of lamprey and their ecosystems. I look forward to seeing it published in *Nature Communications* after minor revisions to the discussion about conodonts.

Lauren Sallan

Reviewer #3:

Remarks to the Author:

GENERAL COMMENTS:

This manuscript describes two very well-preserved large lampreys, each identified as a new species, from the Middle-Late Jurassic Yanliao Biota of North China, dating to approximately 163-158 million years ago. Their very large body size (especially *Yanliaomyzon occisor*, which exceeds 700 mm in length) is far greater than any fossil lamprey yet described and is on par with some of the largest modern anadromous lampreys. This find is very exciting, particularly given the recent study by Miyashita et al. (2021) showing that the ammocoete stage in lampreys is a derived characteristic, and the presumption by previous authors (e.g., Evans et al. 2018; Docker and Potter 2019, attached) that, following the evolution of a radical metamorphosis, the ammocoete larval stage and the post-metamorphic juvenile stage subsequently became well-adapted to their respective environments and the growth potential of each period was maximized, enabling the large body size that now characterizes modern parasitic lampreys. The fact that these hypotheses have now been demonstrated and dated in the fossil record is very exciting. As other authors have pointed out, lampreys have maintained a conserved morphology over hundreds of millions of years, but the evolution of the modern lamprey life history has been more recent and less well understood.

The conclusion that the phylogenetic distribution of the feeding modes suggests that the flesh-eating habit should be ancestral for modern lampreys is likewise novel and interesting. The finding of presumed intestinal contents of several tooth-bearing jaw bones and possibly skull bones of some unidentified bony fishes was particularly interesting, countering previous suggestions that, among modern parasitic lampreys, blood feeding is ancestral (Potter and Hilliard 1987). Potter and Hilliard (1987) argued that blood feeding would be less detrimental to the hosts (and thus, ultimately, to the lampreys which depend on them) in less productive waters. Blood is a renewable resource if the rate of feeding is not excessive relative to the size of the host, and they concluded that flesh feeding came later, with access to smaller but more plentiful coastal fishes (e.g., herrings and other clupeids) or where lampreys travel farther offshore to feed in very productive waters. I think readers would be able to appreciate the significance of your new findings more with a fuller explanation of previous views.

The resemblance to the Southern Hemisphere pouched lamprey was also interesting although, given that dentition in lampreys may evolve convergently in different taxa (e.g., the dentition of *Lampetra ayresii* from western North America is virtually indistinguishable from the dentition of *Lampetra fluviatilis* from western Europe, even though they do not form a monophyletic group), I found the conclusion that modern lampreys originated in the Southern Hemisphere somewhat less definitive. Nevertheless, this is another valuable contribution to our understanding of lamprey evolution.

However, although all these findings are very exciting, I found that the discussion of the manuscript did not highlight them sufficiently. I found the discussion was less concise and less focused than it could have been, which, in my opinion, diluted and distracted from the novel and exciting results. Some suggestions for improvement are below.

Overall, these fossils represent a VERY exciting discovery. With revision, I think the manuscript is suitable for publication in Nature Communications. With a discussion that more concisely highlights the importance of these findings and better places them in the context of what we already know or have surmised about the evolution of lamprey life history type, I think it would be of considerable interest to both a broad audience and lamprey biologists specifically.

SPECIFIC COMMENTS:

Line 4 – Virtually all authorities now recognize lampreys and hagfishes as a monophyletic group, making lampreys and hagfishes equally “the oldest living jawless vertebrates”

Line 61, 62 – What classification are you following? Please indicate. Recent classifications with which I am familiar (e.g., Nelson et al. 2016, *Fishes of the World*, 5th edition) place hagfishes and lampreys

into separate classes (Myxini and Petromyzontida, respectively) and I have not seen *Hyperoartii* used as a synonym for order Petromyzontiformes.

Line 110 – Do modern flesh-eating lampreys have smaller oral discs relative to blood-sucking lampreys (i.e., are oral discs 4.8% and 6.7% of total body length inconsistent with blood-feeding)? Please be explicit.

Line 138-139 – Are you including *Geotria macrostoma* in your enumeration? Presumably yes since you cite Riva-Rossi et al. (2020; citation 27) in this manuscript, but both references that you cite with respect to the gular pouch (Monette and Renaud 2005; Renaud 2011) were published before the revalidation of *Geotria macrostoma*; if you've taken the information directly from Monette and Renaud (2005) and Renaud (2011), it won't be up to date.

Line 188-192 – This paragraph seems insufficient for a broader audience unfamiliar with lamprey biology while also coming across as somewhat superficial for audiences with a familiarity of different lamprey/fish life history types. This is particularly the case for the sentences "In body length, *Yanliaomyzon occisor* falls within modern anadromous lampreys but notably departs from the smaller freshwater residents and non-feeding forms⁵. This implies for the anadromy wherein the higher marine productivity could have provided more feeding opportunities and hence greater growth potential." It has already been stated in this manuscript (Line 101) that these two species are among the largest of modern anadromous lampreys, and the differences in growth potential between anadromous and freshwater fishes (not just lampreys) is generally well known, so the suggestion here that higher marine productivity could have provided more feeding opportunities and hence greater growth potential isn't original. It would seem more appropriate to state that the large body size here is consistent with known larger body size in anadromous fishes compared to their freshwater counterparts or that, for the first time, your findings provide evidence of this predicted transition to anadromy and have been able to date it. Placing your new findings in the context of what is already known or what has already been predicted about the evolution of lamprey life history type would make this more interesting. Also, the term "non-feeding forms" might be misleading to those unfamiliar with non-parasitic lampreys since they are only non-feeding after the onset of metamorphosis.

Line 192-195 – It is not clear here if you are suggesting that *Yanliaomyzon* evolved a triphasic life cycle or rather that it inherited it from the common ancestor that possessed a triphasic life cycle. I think the latter, but the wording could suggest the former.

Line 221 – The meaning of "which should be historical products of the enhancement of the feeding structures" is not clear to me

Line 234-254 – As with Line 188-192, this section seems rather simplistic, neither clearly explaining the "big picture" for a broad audience nor concisely and clearly describing what is novel here for a more expert audience.

EDITORIAL COMMENTS:

The language used sometimes appears too sensational or not quite appropriate, for example:

Line 5, 29 – "bizarre" feeding behavior

Line 6, 29 – "gouging out" tissues from their "victims"

Line 38 – "renovated" the life history strategy

Line 127 – "tricuspid cusps"

Line 179-180 – a large, "monstrous" flesh eater that can even "destroy" the skull of teleost fish

Suggested corrections:

Line 163 – petromyzontiform history (instead of petromyzontiform's history), assuming that Mesozoic lampreys are placed in the same order as extant lampreys. Is that the case?

Typographical error noted:
Line 387 – Gills should be Gill

RESPONSES TO REVIEWER COMMENTS

Reviewer #1 (Remarks to the Author):

This is an outstanding contribution to our knowledge of lamprey phylogeny and biogeography through the integration of all known fossil and virtually all extant lampreys (*Geotria macrostoma* excluded) **(Re: Yes, despite of the differences in the details of the teeth and some ecological features, there is a consensus that *Geotria macrostoma* and *Geotria australis* are most closely related species (relevant refs cited in our study), herein we choose the latter as the representative of this genus as its morphological data is currently more complete than the other)** that feed post-metamorphosis. The discovery of two new fossil lampreys has enabled this breakthrough.

I have a number of comments and suggested changes to improve the manuscript.

The authors refer to the Jurassic era, but shouldn't it be Jurassic Period? **(Re: Rephrased already)**

In the Abstract, the three phases of the life cycle should be specified for clarification. **(Re: We have rephrased this part and specified the three phases of living lampreys' life cycle)**

The word "hosts" at the end of the Abstract and Introduction should be replaced by prey because, given their oral and lingual dentitions, the two newly-described fossil lamprey species were more likely predators rather than parasites. **(Re: Thanks for your suggestion. We have revised the words for lampreys' hosts (for blood-feeders) or preys (for flesh-feeders) in the text.**

However, we referred 'hosts' to the lineage represented by *Mesomyzon* which was estimated to stem from the Early Jurassic and is likely a blood-feeder (referring to ref. 9, our restudy of *Mesomyzon* judged by its body size and the features of its suctorial oral disc). In other words, we think the earliest lampreys with modern-type life cycle should be blood-feeding when other fishes evolved thinned integument (from enameled scales in primitive forms to bony cycloid scales of advanced teleosts))

In the Abstract, the word "the" should precede Pangean supercontinent. **(Re: Corrected.)**

In the Introduction, it is unclear to me what is meant by "simply assembled analogue". This needs to be clarified. **(Re: We referred to their simply assembled teeth, and this part has been**

rephrased)

In the Introduction, the first occurrence of the expression “feeding mechanism” should be replaced by feeding apparatus. **(Re: Yes, we have replaced “feeding mechanism” with “feeding apparatus” in the text)**

In the Introduction, replace “the modern-type life history mode” by “one of the modern-type life history modes”. There are four modes of life in extant lampreys: blood feeders (parasites), flesh feeders (predators), blood and flesh feeders (intermediates), and a presumed carrion feeder (scavenger) (Renaud and Cochran 2019). The full reference is: Renaud C.B. and P.A. Cochran. 2019. Post-metamorphic feeding in lampreys. pp. 247-285. In: Lampreys: Biology, Conservation, and Control. vol. 2, M.F. Docker (ed.), Fish and Fisheries Series 38, Springer Nature B.V. https://doi.org/10.1007/978-94-024-1684-8_3 **(Re: Thanks for providing the reference which has been cited in the revised text. In fact, here we referred to the life cycle (from ammocoete to adult) mode, not living or feeding mode)**

At the end of the Introduction, change “renewed” to “revised” lamprey crown-group phylogeny. **(Re: Corrected)**

Under Systematic Palaeontology, the oral disc size is not admissible as a diagnostic character for the new genus because only one of the two new species has a measurable oral disc (see Supplementary Table 1) **(Re: Indeed, the oral disc of the holotype of *Y. ingensdentis* was partially distorted and cannot be accurately measured, therefore the measurements in S Table 1 have been revised accordingly)**. Notwithstanding, the oral disc is stated to be relatively small (at least in *Y. occisor*); compared to what? **(Re: The ratio of the relative size of the oral disc of *Y. occisor* falls within the ratio of living flesh-feeding lampreys which is larger than that of the typical blood-feeding species, please also check the replies below and the revised text for this issue)** The oral disc size should be quantified and compared. How do you know that the teeth are made of keratin? **(Re: Technically, we failed to determine the nature of the teeth of our fossils, as we conducted the element analyses via electronic scanning, we only detected Si element in the position of the teeth, which suggests that the teeth might have been replaced by the mineral elements from the rock matrix (This implies for some special preservation settings. Notably, some flying dinosaurs with membranous wings were discovered from the same layers). For now, we can just take them as the analogues of keratinous teeth in living lampreys.)** It is

sufficient to say posterior disc teeth lacking (i.e., to include and “posterior” circumoral is implicit). I agree that the anterior and lateral circumoral teeth are spatulate in shape, but cannot understand what is meant by “and each bearing three convergent blades” **(Re: Admittedly, the description was not clear enough. Each circumoral tooth looks like a trihedron that bears three edges of different lengths. We have replaced the reconstructions of the oral discs and dentition of our fossil lampreys (Fig. 2d, f) with high-quality computer-generated images that will illustrate the anatomy much better).**

Figs 2d and f appear to be mislabeled **(Re: The mislabelings have been corrected; as for the description of the circumoral teeth, we meant that each tooth looks like a trihedron with the tip pointing to the direction of the oral aperture; we have revised Fig. 2d, f to illustrate more clearly the morphology of these teeth)**

According to the species’ diagnoses, Fig. 2d would correspond to a drawing of *Y. ingensdentis* and Fig. 2f to a drawing *Y. occisor* **(Re: Yes, these figs were mislabeled and the figure legend has been fixed)**. In the figure caption, however, both Figs 2d and f are stated to correspond to *Y. occisor*, respectively, as a photograph and a restoration **(Re: Corrected)**.

I disagree with the limits of the anterior and lateral fields delineated in Fig. 2g. According to Renaud (2011: 7) (reference 11 in the manuscript), the boundaries are set by the anteriormost edge of the supraoral lamina and a line drawn through the apices of the two lateralmost cusps on the infraoral lamina. These limits, as defined by Renaud (2011), have repeatedly been used in the literature for counts of lamprey teeth, including those of *Geotria*, which the authors state resemble those of *Yanliaomyzon*. **(Re: Yes, we have corrected the limits of the anterior and lateral fields of oral disc teeth in Fig. 2g according to your suggestions and relevant literatures)**

In the diagnosis for *Y. occisor*, the tail region is stated to measure about one fourth of the total body length, but according to Supplementary Table 1, this rather equals 38.7%! (i.e., $a-C/TL = 270.7/700 = 38.7\% > 25.0\%$). **(Re: Yes, indeed! Thank you for pointing out these problems. We measured again the holotype and corrected some measurements. Now we calculated that the tail length occupies ca. 28% of total body length, which is still much shorter than that in *Y. ingensdentis* (slightly more than 40 %)).**

In the etymology for *Y. ingensdentis*, I would change rasping piston to gouging piston, because the latter is the action of a flesh feeder, while the former is the action of a blood feeder. **(Re: Yes, we have replaced ‘rasping’ piston with ‘gouging’ piston for this flesh-eating lamprey)** In the diagnosis of *Y. ingensdentis*, the dorsal fin is said to be contiguous with the high and large anterior caudal fin, but I believe the interpretation of the fins is incorrect. Rather, the low first dorsal fin is contiguous with the high second dorsal fin and the latter is continuous with the caudal fin. **(Re: Yes, we get your points. Actually, different from the conventional terminology for lampreys’ fins, we (Wu et al., 2021, [p.1298 in ref. 8 in current study]; Janvier, 2008 [p.1051 in ref. 2 in current study]) ever tentatively homologized the ‘posterior dorsal fin’ with the anterior part of dorsal lobe of the caudal fin that is mostly located behind the level of the cloaca (In fishes, this opening marks the boundary of the abdominal and caudal regions). Additionally, regarding the distribution of the cartilaginous radials of the fins, we also noted that those of the ‘posterior dorsal fin’ and the caudal fin are always continuous in the vast majority of living lampreys (with the exception of *Geotria*) (Renaud, 2011), probably reflecting a closer relation between these two parts than that between the anterior (first) and the conventionally-termed posterior (second) dorsal fins. Especially when we have to compare lampreys with other early vertebrates (including those fossil jawless vertebrates in Devonian, e.g., the ‘naked anaspids’ (see Janvier, 2008, ref. 2 in current study)) from an evolutionary perspective, we will encounter the problem of the homology of the dorsal median fins of early vertebrates. Our hypothesis might be an alternative solution.)**

In the Description and comparison, it is stated that *Y. occisor* is > 700 mm, but Supplementary Table 1 indicates exactly 700 mm, and therefore, the symbol greater than (i.e., >) should be deleted. This new species is said to be only slightly smaller than *P. marinus* (1,200 mm) and *E. tridentatus* (850 mm), but in this group should also be included *G. australis* (788 mm) and *L. camtschaticum* (790 mm) (see Renaud and Cochran 2019). Given these data, the qualifier “slightly” is incorrect and should be removed. **(Re: Yes, we have measured again the fossils and included all these large**

living species in the text and deleted the qualifier “slightly” . The ref. you referred to has been cited in the revised text.)

The relatively small oral discs of *Y. occisor* and *Y. ingensdentes* are stated to be respectively, 4.8% and 6.7% of total length, but the former is correctly 4.7% and I can't find the data to derive the latter because Supplementary Table 1 does not have a value for the disc length of *Y. ingensdentes* (**Re: Thank you for pointing out these problems. We rechecked the fossils and fixed the measurements of Supplementary Table I. The oral disc of the holotype of *Y. ingensdentes* was partially distorted and cannot be accurately measured, therefore the measurements in S Table 1 was revised accordingly. We also rechecked the data of *Y. occisor* and revised some measurements of this species**). To state that the oral disc length to total body length of the two *Yanliaomyzon* species approach the condition of modern flesh-eating lampreys is not informative without providing the data. The data in Potter and Hilliard (1987) give a range for this ratio of 2.7–6.3% for flesh feeders (and specifically, 2.7–3.2% in *Geotria*) and 5.8–11.8% for blood feeders. (**Re: We measured the oral disc size of *Yanliaomyzon occisor* again and corrected the numbers in the text and S Table 1**)

As mentioned above, to state that posterior disc teeth are missing is sufficient because posterior circumoral teeth are by definition disc teeth. As also mentioned above, the meaning of the longest of the three converging blades facing slightly sideways is unclear to me (**Re: We have revised Fig. 2d, f and the description to show more clearly the structure of the circumoral teeth**).

The expression tricuspid cusps is nonsensical (a tooth or a lamina may be tricuspid, but not a cusp). It would make more sense to say The tips of these three cusps (**Re: Thanks, the text here has been corrected**)

The prebranchial region is stated to be longer than the branchial apparatus in the two fossil lampreys; and this is certainly borne out in *Y. ingensdentes* (Supplementary Table 1: $d-B1 = 42$ versus $B1-B7 = 12$), but neither of these morphometric characters is recorded in *Y. occisor*. (**Re: Indeed, we missed some values for *Y. occisor* in the table. Actually, for these values we measured the**

paratype (IVPP V 18956B) of *Y. occisor* (Fig. 1c, d) where the impression of the head and branchial region was preserved but not in the holotype. The numbers now have been added in Supplementary Table 1)

The eyes are stated to be medium-sized, but while a value of 4 is recorded in Supplementary Table 1 for *Y. ingensdentis*, none is given for *Y. occisor*. (Re: Indeed, for this value we measured the paratype of *Y. occisor* (Fig. 1c, d) and added the value in Supplementary Table 1)

The gular pouch is supposed to present in Fig. 2c and in Extended Data Fig. 2k, but this is certainly not evident to me (Re: Yes, this structure is not completely preserved. However, according to the part (Fig. 2a) and counterpart (Fig. 2c, Extended Data Fig. 2k, with the gular pouch better preserved than in Fig. 2a)) of the specimen, we saw a depression with a curving lateral edge (Fig. 2a, b) that matches the pouch-like protrusion (comparable to the preservation state of the gular pouch in *Y. occisor* (see Fig. 1b)). Despite lacking the ventral part, the remaining part of this protrusion shows a pouch-like profile (compare to the preserved gular pouch of *Y. occisor* [Fig. 1b]) and bears some wrinkles produced by the tissues of gular pouch (pointed by the white arrow in Fig. 2c). It is present in Fig. 2e, but only indicated with the abbreviation g and is missing the associated letter p, as defined in the anatomical abbreviations (Re: We have rechecked the material of Fig. 2e and confirmed the large structure with a round ventral edge and three tips which was delineated as the ‘transverse lingual lamina’ [‘tl’]; we have also checked and added the labeling of the gular pouch (gp) in the revised Fig. 2c).

According to references 11 and 22, there are in fact 11 (not nine) living lamprey species with a gular pouch, and six (not five) of these are anadromous (Re: Yes, indeed, we have checked and rephrased the text here). However, you also need to add *Geotria macrostoma*; making 12 species with a gular pouch, seven of which are anadromous. The reference for *G. macrostoma* is: Baker, C.F., C. Riva Rossi, P. Quiroga, E. White, P. Williams, J. Kitson, C.M. Bice, C.B. Renaud, I. Potter, F.J. Neira, and C. Baigún. 2021. Morphometric and physical characteristics distinguishing adult Patagonian lamprey, *Geotria macrostoma* from the pouched lamprey, *Geotria australis*. PLoS ONE 16(5): e0250601. <https://doi.org/10.1371/journal.pone.0250601> (Re: Yes, we have included this

reference in the text and revised the text therein)

Instead of Before the cloaca, a long preanal skin fold ... , I suggest A long precloacal skin fold ...

(Revised)

I suggest intestinal tract rather than intestinal duct. **(Revised)**

Under Feeding mode, life history and swimming performance of Yanliaomyzon change stomach to intestinal or gut content.**(Revised)**

The sentence This implies for the anadromy wherein the higher marine productivity ... must be missing some words because it makes no sense.**(Re: The sentence has been rephrased.)**

As suggested above, the preanal skin fold should be called precloacal skin fold. The long-based dorsal fin and ribbon-like precloacal skin fold considerably extend rather than extends.**(Re: We have revised this part of description according to your suggestion)**

Under The evolution of lampreys' feeding structures and its implications, I do not understand what is meant by hitch-hiking on other delivers. I suspect the word delivers is not what you mean.**(Revised)**

I would term the radical metamorphic phase, a stage rather than a phase.**(Corrected)**

Under The early biogeographical history of modern lampreys, shouldn' t Late Jurassic and Late Cretaceous be periods instead of eras? **(Revised)**

I believe that the word promoted in relation to trans-equatorial dispersal is inappropriate and should maybe be replaced by led to.**(Revised)**

I question the relevance of reference no. 48 in relation to major fish hosts of modern lampreys because in their revision of post-metamorphic feeding in lampreys, Renaud and Cochran (2019) do not mention the genus Trisopterus as a host of lampreys.**(Re: Thank you for reminding us of the**

reference and the book (volume 2 of ‘Lampreys: Biology, Conservation and Control’) that included this paper. We have gone through carefully these references and agree with you that we should exclude *Trisopterus* from list of the host victims of lampreys.)

Under Anatomical abbreviations, the adf, anterior dorsal fin should be called first dorsal fin and the pdf, posterior dorsal fin should be called second dorsal fin. How can the posterior dorsal fin be synonymous with the anterior part of the caudal fin? I note that in lines 237-240 of the Supplementary Information, the authors have justified their position in regards to dorsal and caudal fin definitions, but to me it remains unsatisfactory. While in Petromyzontidae the second dorsal fin (all genera minus *Ichthyomyzon*) or posterior lobe of the single dorsal fin (*Ichthyomyzon*) is confluent with the caudal fin, *Geotria australis* is known to have a distinct second or posterior dorsal fin, separated by an interspace from the caudal fin **(Re: Thanks. The ‘confluence’ between the ‘second dorsal fin’ and the caudal fin in stem lampreys, as well as in the majority of living lampreys suggests that this state should be ancestral of crown lamprey lineages. *Geotria*’s disjunction of these two parts is a derived feature). (For this issue, we responded to your comments above. We reinterpreted the conventionally-termed ‘posterior dorsal fin’ as the anterior part of the caudal fin according to its relative position to the cloaca and a broader comparison to early jawless vertebrates from an evolutionary perspective).**

Note that in reference no. 28, the second author is Gill, not Gills.**(Corrected)**

Note that in reference no. 34, the word Environments should have its first letter in lower case. **(Corrected)**

Note that in reference no. 42, the word Pacific should have its first letter in upper case. **(Corrected)**

Figs 3 and 4 contain much valuable information, but are too small for readers to be able to discern many of the details and should be increased in size. In regards to Fig. 4, it is unclear to me to which fossil taxa the four red spots refer to. Additionally, the distinction between the light blue and dark blue shadows is indiscernible. **(Re: Thank you for your suggestions. We have reorganized Figs 3 and 4, increasing the size and the color contrast between different units. Also, we have added panels in Fig. 4 to illustrate the results of the palaeobiogeographical analysis and show the**

possible ancestral geographical distributions for some relevant nodes, e.g., that of the crown-group lampreys.)

The title of Supplementary Table 1 is misleading and confusing. A meristic measurement is a misnomer. The only meristic character it contains is the number of trunk myomeres and this has only been determined in *Mesomyzon mengae*. I suggest deleting Meristic from the title and simply starting with Measurements and removing the column number of trunk myomeres. I presume that all measurements are in mm, but this should be specified. **(Re: Yes, we have revised the title of the table, added ‘all measurements are in mm’ and deleted the column of the number of trunk myomeres)**

For completeness and balance of opinion, I strongly recommend the inclusion of two recent papers into the Discussion; namely, Brownstein and Near (2023) and Mallatt (2022). The complete references are: Brownstein, C.D. and T.J. Near. 2023. Phylogenetics and the Cenozoic radiation of lampreys. *Current Biology* 33: 1-8; Mallatt, J. 2022. Vertebrate origins are informed by larval lampreys (ammocoetes): a response to Miyashita et al., 2021. *Zoological Journal of the Linnean Society* XX: 1-35. **(Re: Thank you for providing the information of these important literatures. We have included these refs in the revised MS and rewritten the relevant part of the text. We noticed that although Brownstein, C.D. and T.J. Near. (2023) highlighted the Cenozoic radiation of the lampreys in the Northern Hemisphere, their topology of the phylogeny and the timing of the node of the crown-group lampreys are radically different from those in our study.)**

Claude B. Renaud

Reviewer #2 (Remarks to the Author):

This paper presents a radical revision of the evolutionary and ecological history of lampreys based on an exceptionally preserved fossil from the Jurassic. Prior fossils had suggested that the feeding morphology of lamprey was highly constrained for the last 360 million years with the exception of the pouched lamprey, and that key features like ammocete larvae evolved at some unknown time.

The present study uses new data from a Jurassic fossil to support an abrupt change in lamprey ecology around the Mesozoic origin of the crown, with flesh-eating as the ancestral state and greater diversity in feeding morphology than anticipated. The study is very well written and wide ranging and thorough in its methods, implications for the evolution of fishes, and in the number of factors considered as drivers for the diversification of lamprey.

However, the one, somewhat glaring, omission is a lack of consideration of conodonts, which recently have been assigned to the cyclostome stem in other studies of fossil cyclostomes. Conodonts were flesh-eaters with feeding apparatus that sometimes resemble the novel plates of the new lamprey. They also finally went extinct at the end-Triassic. It seems possible that the extinction of sometimes large-bodied (>1m), pelagic, shearing-biting conodonts may have opened space for the previously non-biting lamprey (although conodonts were never freshwater). Conodonts are often ignored in evolutionary studies of fishes, but in this case there could be a real influence on the timing of lamprey diversification and the change in their ecology proposed here. I would like to see some discussion of this possibility. **(Re: Thank you very much for providing this enlightening point. You are definitely right, so we have to fix it, as the conodonts and lampreys ever cooccurred for such a long geological history, are so closely related phylogenetically and are very similar in the overall morphology (body plan) and the functional anatomy of the oral apparatus (lingual lamina system), conodonts were probably the ecological competitors when they coexisted with lampreys and their extinction in the latest Triassic must have arguably 'released' new ecological niches for lampreys, which might have inspired the evolution of lampreys during the following Jurassic period, a crucial period of lampreys' history as highlighted in our study. Following your suggestions, we have added some comments and cited two relevant references in the text. The new refs are: (1) Ginot, S. and Goude mand N. Conodont size, trophic level, and the evolution of platform elements. *Paleobiology*, 1-11 (2019). doi: 10.1017/pab.2019.19**

(2) Iannicelli, M. Solving the mystery of endless life between conodonts and lampreys, plus a reason for final extinction of the conodonts. *J Oceanogr Mar Res S1*: 001(2017). doi: 10.4172/2572-3103.S1-001)

In any case, this discovery and resultant investigation raises a number of interesting possibilities for

and questions about Mesozoic fish evolution, which will lead to further reconsideration of the record of lamprey and their ecosystems. I look forward to seeing it published in Nature Communications after minor revisions to the discussion about conodonts.

Lauren Sallan

Reviewer #3 (Remarks to the Author):

GENERAL COMMENTS:

This manuscript describes two very well-preserved large lampreys, each identified as a new species, from the Middle-Late Jurassic Yanliao Biota of North China, dating to approximately 163-158 million years ago. Their very large body size (especially *Yanliaomyzon occisor*, which exceeds 700 mm in length) is far greater than any fossil lamprey yet described and is on par with some of the largest modern anadromous lampreys. This find is very exciting, particularly given the recent study by Miyashita et al. (2021) showing that the ammocoete stage in lampreys is a derived characteristic, and the presumption by previous authors (e.g., Evans et al. 2018; Docker and Potter 2019, attached) that, following the evolution of a radical metamorphosis, the ammocoete larval stage and the post-metamorphic juvenile stage subsequently became well-adapted to their respective environments and the growth potential of each period was maximized, enabling the large body size that now characterizes modern parasitic lampreys. The fact that these hypotheses have now been demonstrated and dated in the fossil record is very exciting. As other authors have pointed out, lampreys have maintained a conserved morphology over hundreds of millions of years, but the evolution of the modern lamprey life history has been more recent and less well understood. **(Re: We have revised the text and figures by improving the quality of the reconstructions of our fossil lampreys' feeding apparatus (Fig. 2d, f), adding the results of ancestral state and biogeographical reconstructions and mapping them on the renewed Figs. 3 & 4, please check the revised MS, which should make our points clearer.)**

The conclusion that the phylogenetic distribution of the feeding modes suggests that the flesh-eating habit should be ancestral for modern lampreys is likewise novel and interesting. The finding of presumed intestinal contents of several tooth-bearing jaw bones and possibly skull bones of some

unidentified bony fishes was particularly interesting, countering previous suggestions that, among modern parasitic lampreys, blood feeding is ancestral (Potter and Hilliard 1987). Potter and Hilliard (1987) argued that blood feeding would be less detrimental to the hosts (and thus, ultimately, to the lampreys which depend on them) in less productive waters. Blood is a renewable resource if the rate of feeding is not excessive relative to the size of the host, and they concluded that flesh feeding came later, with access to smaller but more plentiful coastal fishes (e.g., herrings and other clupeids) or where lampreys travel farther offshore to feed in very productive waters. I think readers would be able to appreciate the significance of your new findings more with a fuller explanation of previous views. **(Re: Thank you for your suggestions. We have reorganized and rephrased the text by adding the results of our ancestral state reconstructions of the feeding apparatus and mapping them on a renewed Fig. 3, please check the revised MS)**

The resemblance to the Southern Hemisphere pouched lamprey was also interesting although, given that dentition in lampreys may evolve convergently in different taxa (e.g., the dentition of *Lampetra ayresii* from western North American is virtually indistinguishable from the dentition of *Lampetra fluviatilis* from western Europe, even though they do not form a monophyletic group), I found the conclusion that modern lampreys originated in the Southern Hemisphere somewhat less definitive. Nevertheless, this is another valuable contribution to our understanding of lamprey evolution. **(Re: We have reorganized and rephrased the text by adding the results of our biogeographical analysis and mapping them on the lamprey timetree in the revised Fig. 4, please check the revised MS)**

However, although all these findings are very exciting, I found that the discussion of the manuscript did not highlight them sufficiently. I found the discussion was less concise and less focused than it could have been, which, in my opinion, diluted and distracted from the novel and exciting results. Some suggestions for improvement are below. **(Re: We have reorganized and rephrased the section of discussion accordingly, please see the revised MS)**

Overall, these fossils represent a VERY exciting discovery. With revision, I think the manuscript is suitable for publication in Nature Communications. With a discussion that more concisely highlights the importance of these findings and better places them in the context of what we already know or have surmised about the evolution of lamprey life history type, I think it would be of considerable interest to both a broad audience and lamprey biologists specifically. **(Re: We have rephrased the text accordingly, please see the revised MS)**

SPECIFIC COMMENTS:

Line 4 – Virtually all authorities now recognize lampreys and hagfishes as a monophyletic group, making lampreys and hagfishes equally “the oldest living jawless vertebrates”**(Re: Revised. In the former version, we referred to the known fossil records of these two jawless groups, in which lampreys have older fossils than hagfishes)**

Line 61, 62 – What classification are you following? Please indicate. Recent classifications with which I am familiar (e.g., Nelson et al. 2016, Fishes of the World, 5th edition) place hagfishes and lampreys into separate classes (Myxini and Petromyzontida, respectively) and I have not seen Hyperoartii used as a synonym for order Petromyzontiformes. **(Corrected.)**

Line 110 – Do modern flesh-eating lampreys have smaller oral discs relative to blood-sucking lampreys (i.e., are oral discs 4.8% and 6.7% of total body length inconsistent with blood-feeding)? Please be explicit. **(Re: The text here has been revised according to your suggestion. According to the study of Potter and Hilliard, 1987 (ref.10 of current study), this ratio is 2.7–6.3% for flesh feeders, and 5.8–11.8% for blood feeders, reflecting feeding mode of the blood feeders that require the attachment on the host victims for longer time than the flesh feeders)**

Line 138-139 – Are you including *Geotria macrostoma* in your enumeration? Presumably yes since you cite Riva-Rossi et al. (2020; citation 27) in this manuscript, but both references that you cite with respect to the gular pouch (Monette and Renaud 2005; Renaud 2011) were published before the revalidation of *Geotria macrostoma*; if you’ve taken the information directly from

Monette and Renaud (2005) and Renaud (2011), it won't be up to date. (**Re: Thank you for reminding us of this. We have fixed it and also added a reference--Baker, C.F., C. Riva Rossi, P. Quiroga, E. White, P. Williams, J. Kitson, C.M. Bice, C.B. Renaud, I. Potter, F.J. Neira, and C. Baig ú n. 2021. Morphometric and physical characteristics distinguishing adult Patagonian lamprey, *Geotria macrostoma* from the pouched lamprey, *Geotria australis*. PLoS ONE 16(5): e0250601**)

Line 188-192 - This paragraph seems insufficient for a broader audience unfamiliar with lamprey biology while also coming across as somewhat superficial for audiences with a familiarity of different lamprey/fish life history types. This is particularly the case for the sentences "In body length, *Yanliaomyzon occisor* falls within modern anadromous lampreys but notably departs from the smaller freshwater residents and non-feeding forms⁵. This implies for the anadromy wherein the higher marine productivity could have provided more feeding opportunities and hence greater growth potential." It has already been stated in this manuscript (Line 101) that these two species are among the largest of modern anadromous lampreys, and the differences in growth potential between anadromous and freshwater fishes (not just lampreys) is generally well known, so the suggestion here that higher marine productivity could have provided more feeding opportunities and hence greater growth potential isn't original (**Re: Yes, we agree with you. There are sufficient evidences that those species and the populations that feed in marine ecosystem. We have added a reference-- Renaud CB and Cochran PA. Post-metamorphic Feeding in Lampreys. In Docker, M. F.ed. Lampreys: Biology, Conservation and Control, p.247-286 (Springer,2019)).**)

It would seem more appropriate to state that the large body size here is consistent with known larger body size in anadromous fishes compared to their freshwater counterparts or that, for the first time, your findings provide evidence of this predicted transition to anadromy and have been able to date it. (**Re: Yes, we have rephrased this part**) Placing your new findings in the context of what is already known or what has already been predicted about the evolution of lamprey life history type would make this more interesting (**Re: Indeed, we have rewritten this part and noted the he conventional wisdom that the living lampreys should be ancestrally blood-feeders as the dentitional pattern represented by *Ichthyomyzon* species was always held as the primitive state**)

for living lampreys, see Potter & Hilliard, 1987; Hardisty 2015; Renaud & Cochran, 2019). Also, the term “non-feeding forms” might be misleading to those unfamiliar with non-parasitic lampreys since they are only non-feeding after the onset of metamorphosis (**Re: We have made it explicit that here we refer to the non-feeding habit during the post-metamorphosis stage**).

Line 192-195 – It is not clear here if you are suggesting that *Yanliaomyzon* evolved a triphasic life cycle or rather that it inherited it from the common ancestor that possessed a triphasic life cycle. I think the latter, but the wording could suggest the former. (**Re: We have revised this part and made it clear that ‘*Yanliaomyzon* species could likely have inherited such a life history strategy from the common ancestor’**)

Line 221 – The meaning of “which should be historical products of the enhancement of the feeding structures” is not clear to me (**Re: We have rephrased this part**)

– As with Line 188-192, this section seems rather simplistic, neither clearly explaining the “big picture” for a broad audience nor concisely and clearly describing what is novel here for a more expert audience. (**Re: We have tried to improve the expression of the co-evolutionary interaction of the lampreys and their host (prey) victims, please check the revised MS. The text in this part has been rephrased. Also, we have refreshed the text figures to help the readers understand our points**)

Reviewers' Comments:

Reviewer #1:

Remarks to the Author:

The authors have satisfactorily answered the majority of my concerns. There are still a few issues that I would like them to correct and new ones have cropped up. A copy editor will need to carefully go through the revised version because, for example, some spaces between words no longer exist (l. 102), commas occur where they shouldn't (l. 68), spaces need to be removed (l. 68), references should start on a new line (l. 476).

Lines 66-67: Please remove "and posterior circumoral" because if there are no posterior disc teeth there can't be any posterior circumoral teeth either.

The authors use stage and phase indiscriminately. I use phase as a subset of stage (e.g., feeding phase of the adult stage). I have no problem if the authors choose to refer to a three-phase lamprey life cycle as comprising the ammocoete, metamorphic, and adult, but they need to be consistent. It is confusing to readers if they call the three-phased lamprey life cycle, ammocoete, metamorphic, and adult stages.

Line 84 and Fig. 2f: The two cusps flanking the two central cusps of the supraoral lamina in *Y. occisor* do not appear smaller to me! They appear of similar size to the central cusps.

Lines 85 and 99: Remove "body very slender with" and "body relatively deep with the", respectively, because no measurement of the body depth was taken. Furthermore, remove "BD?", presumably body depth, in line 685 of Supplementary Table 1 because this morphometric does not appear in the table.

Line 99: Add the word slightly before "more than 40% of total body length."

Line 106: Add the common name Arctic lamprey before *Lethenteron camtschaticum*.

Line 301: Lampreys do not use their oral disc to move sand, but use it to move stones to line the periphery of a semi-circular redd. They move sand using their tail to create a central depression in the redd into which the fertilized eggs fall into.

Claude B. Renaud

Reviewer #2:

Remarks to the Author:

The authors have produced a well-rounded revision that addresses all the comments of the reviewers. I really appreciate the expanded discussion about the ecological factors behind the evolution of modern lamprey, including the potential relationship between the loss of enameloid in teleost scales and the evolution of parasitism. My main comment is that the writing in this new discussion section needs additional polishing, as there are multiple grammatical errors (tense, extra spaces, etc). After this is addressed, I believe this will be a valuable and surprising contribution to the literature on lamprey evolution and the evolution of fish biodiversity, worthy of publication in *Nature Communication*.

Lauren Sallan

Reviewer #3:

Remarks to the Author:

GENERAL COMMENTS:

Thank you for your detailed and thoughtful responses to the excellent feedback you received from the three reviewers and the substantial revisions that you made to the manuscript based on this feedback. I think you have addressed most of the comments related to the manuscript's content, although I bring up a few follow-up comments. I still think that a tighter, more concise discussion would better convey and highlight the novel and exciting results of this paper, but I would not like to see publication of this work held up by several rounds of editing. However, I have pointed out some minor

grammatical errors or editorial issues that I think can be easily corrected.

SPECIFIC COMMENTS REGARDING CONTENT

Line 4 – By saying “the exemplar lineage,” does that exclude hagfishes? Would “one of two lineages” be more appropriate?

Line 22 – What is meant by recent?

Line 27 – As with Line 4, I’m not sure that lampreys are “the” representative lineage

Line 31 – I would say: “in some cases, where they are non-native, they bring tremendous loss for fishery economy”. In their native ranges (i.e., everywhere but in the Laurentian Great Lakes), lampreys are part of a healthy ecosystem.

Line 50 – Include scientific name on first mention?

Line 60 – What classification are you following for Class Cyclostomi? Please indicate. As indicated previously, recent classifications with which I am familiar (e.g., Nelson et al. 2016, *Fishes of the World*, 5th edition) place hagfishes and lampreys into separate classes (Myxini and Petromyzontida, respectively). I recognize that not all authorities use the same classification, but please specify which recent and accepted classification you are following.

Line 253 – Although *Yanliaomyzon occisor* appears to have been anadromous, I’m not sure that I’m convinced from this alone that “living lampreys were probably ancestrally anadromous.” For example, I’m not sure that there might not have already been life history diversity in “pre-modern” lampreys. Also, it would be useful to specify why you suggest that anadromy was ancestral rather than a purely marine ancestry (see comment on Line 326-332).

Line 320-323 – I’m not sure that there is evidence that conodonts were the main competitor to lampreys (i.e., that they occupied the same ecological niche to the exclusion of many other jawless fishes). Furthermore, lampreys are no more closely related to conodonts than they are to other non-cyclostome vertebrates (i.e., they last shared a common ancestor with conodonts at the same time they last shared a common ancestor with the other non-cyclostome vertebrates).

Line 326-332 – Interesting suggestion that the larger body size and hydromechanical advantage provided by fin morphology permitted upstream migration and the evolution of anadromy rather than vice versa. This is somewhat lost in what I think is still a longer, less focused discussion than it could/should be.

Reference 4 – It would be more useful if the specific chapter within this edited book is cited. It looks like some references to this source come from the introductory chapter (Docker et al. 2015, *Introduction: A Surfeit of Lampreys*, pp 1–34) while others (e.g., Line 302 come from Johnson et al. 2015, *Reproductive Ecology of Lampreys*, pp 265–303)

References 7, 14 – What are the title of these chapters?

EDITORIAL CORRECTIONS:

Line 12, 38 – kin, not kins

Line 15 – the Jurassic period (or the Jurassic)

Line 29 – “sucking blood” and “gouging out tissues” still seems a bit colorful, but I’ll leave that to you

and the editor

Line 29, 54, and elsewhere – prey, not preys

Line 31 – the aquatic ecosystem, the fishery economy

Line 39, 41, and elsewhere – be consistent in spelling of ammocoete

Line 252 – “which implies that this large lamprey would have been anadromous” would sound better

Line 252 – phylogenetic

Line 263 – with high hydromechanical efficiency

Lines 276, 336 – “epic odyssey” and “monstrous” still a bit colorful

Line 277 – acting as

Line 293, 304 – armour (not armours)

Line 309 – “A food-driven tipping point for the prepared” sounds rather awkward

Line 313-319 – a rather long, complicated sentence

Line 319 – disappearance would sound better than vanishment

Line 322 – went extinct

Line 324 – delete “,” after with

Line 367 – the southern and northern Pacific Ocean

Line 375 – have led

REVIEWER COMMENTS

Reviewer #1 (Remarks to the Author):

The authors have satisfactorily answered the majority of my concerns. There are still a few issues that I would like them to correct and new ones have cropped up. A copy editor will need to carefully go through the revised version because, for example, some spaces between words no longer exist (l. 102)(**Re: Corrected**), commas occur where they shouldn't (l. 68) (**Re: Corrected**), spaces need to be removed (l. 68) (**Re: Corrected**), references should start on a new line (l. 476) (**Re: Corrected, thank you very much**).

Lines 66-67: Please remove "and posterior circumoral" because if there are no posterior disc teeth there can't be any posterior circumoral teeth either (**Re: Revised accordingly**).

The authors use stage and phase indiscriminately. I use phase as a subset of stage (e.g., feeding phase of the adult stage). I have no problem if the authors choose to refer to a three-phase lamprey life cycle as comprising the ammocoete, metamorphic, and adult, but they need to be consistent. It is confusing to readers if they call the three-phased lamprey life cycle, ammocoete, metamorphic, and adult stages. (**Re: Yes, indeed. We revised and made the usages consistent in the text**)

Line 84 and Fig. 2f: The two cusps flanking the two central cusps of the supraoral lamina in *Y. occisor* do not appear smaller to me! They appear of similar size to the central cusps. (**Re: We refer to the base of the central cusps and flanking cusps, with that of the former being larger than the that of the latter**)

Lines 85 and 99: Remove "body very slender with" and "body relatively deep with the", respectively, because no measurement of the body depth was taken (**Re: Revised**). Furthermore, remove "BD?", presumably body depth, in line 685 of Supplementary Table 1 because this morphometric does not appear in the table. (**Re: Revised**)

Line 99: Add the word slightly before "more than 40% of total body length." (**Re: Added**)

Line 106: Add the common name Arctic lamprey before *Lethenteron camtschaticum*. (**Re: Added**)

Line 301: Lampreys do not use their oral disc to move sand, but use it to move stones to line the periphery of a semi-circular redd. They move sand using their tail to create a central depression in the redd into which the fertilized eggs fall into. (**Re: Thank you very much for reminding these. We have corrected the text**)

Claude B. Renaud

Reviewer #2 (Remarks to the Author):

The authors have produced a well-rounded revision that addresses all the comments of the reviewers. I really appreciate the expanded discussion about the ecological factors behind the evolution of modern lamprey, including the potential relationship between the loss of enameloid in teleost scales and the evolution of parasitism. My main comment is that the writing in this new discussion section needs additional polishing, as there are multiple grammatical errors (tense, extra spaces, etc) (**Re: Thank you. We have checked and revised accordingly**). After this is

addressed, I believe this will be a valuable and surprising contribution to the literature on lamprey evolution and the evolution of fish biodiversity, worthy of publication in Nature Communication.

Lauren Sallan

Reviewer #3 (Remarks to the Author):

GENERAL COMMENTS:

Thank you for your detailed and thoughtful responses to the excellent feedback you received from the three reviewers and the substantial revisions that you made to the manuscript based on this feedback. I think you have addressed most of the comments related to the manuscript's content, although I bring up a few follow-up comments. I still think that a tighter, more concise discussion would better convey and highlight the novel and exciting results of this paper (**Re: Thank you. We have condensed some parts of the discussion section, hopefully it is better now**), but I would not like to see publication of this work held up by several rounds of editing. However, I have pointed out some minor grammatical errors or editorial issues that I think can be easily corrected (**Re: Indeed, we have checked and corrected**).

SPECIFIC COMMENTS REGARDING CONTENT

Line 4 – By saying “the exemplar lineage,” does that exclude hagfishes? Would “one of two lineages” be more appropriate? (**Re: Revised**)

Line 22 – What is meant by recent? (**Re: Revised. We have specified it as ‘post-Cretaceous’**)

Line 27 – As with Line 4, I'm not sure that lampreys are “the” representative lineage (**Re: Revised**)

Line 31 – I would say: “in some cases, where they are non-native, they bring tremendous loss for fishery economy”. In their native ranges (i.e., everywhere but in the Laurentian Great Lakes), lampreys are part of a healthy ecosystem. (**Re: Thank you for reminding this. The text has been revised accordingly**)

Line 50 – Include scientific name on first mention? (**Re: Yes, the scientific name of the pouched lamprey is now included**)

Line 60 – What classification are you following for Class Cyclostomi? Please indicate. As indicated previously, recent classifications with which I am familiar (e.g., Nelson et al. 2016, Fishes of the World, 5th edition) place hagfishes and lampreys into separate classes (Myxini and Petromyzontida, respectively). I recognize that not all authorities use the same classification, but please specify which recent and accepted classification you are following. (**Re: Revised. To avoid the confusion, we deleted “Class Cyclostomi”, as the taxonomy of the crown-group**

vertebrates is beyond the scope of current study)

Line 253 – Although *Yanliaomyzon occisor* appears to have been anadromous, I'm not sure that I'm convinced from this alone that “living lampreys were probably ancestrally anadromous.” For example, I'm not sure that there might not have already been life history diversity in “pre-modern” lampreys. Also, it would be useful to specify why you suggest that anadromy was ancestral rather than a purely marine ancestry (see comment on Line 326-332). **(Re: Yes, you are right, this point might be somehow lost in the lengthy discussion, which is now condensed. The inference of the anadromy of the node of the crown-group lampreys is based on the topology (Fig. 3) and the distribution of this trait in current phylogenetic framework. Given that *Yanliaomyzon occisor* and the earliest diverged crown lineage *Geotria australis* are all anadromous, the node of the crown-group (the theoretical ancestor of the crown-group lampreys) was inferred as an anadromous form. Indeed, we should do an ancestral trait reconstruction for this node, just like the reconstruction of the feeding (trait) habit. However, the emphasis of this study is the feeding biology, we did not do that for now.)**

Line 320-323 – I'm not sure that there is evidence that conodonts were the main competitor to lampreys (i.e., that they occupied the same ecological niche to the exclusion of many other jawless fishes). Furthermore, lampreys are no more closely related to conodonts than they are to other non-cyclostome vertebrates (i.e., they last shared a common ancestor with conodonts at the same time they last shared a common ancestor with the other non-cyclostome vertebrates). **(Re: Indeed, the phylogenetical status of the conodonts is disputed. To avoid the confusion, we have deleted the ‘phylogenetically closely related’ in the text.**

As for the ecological relationships between conodonts and lampreys, although Goudemand et al. (2011), in addition to refs. (39, 43) cited in the original version of our MS, compared the feeding apparatus of the conodonts and pouched lamprey *Geotria australis* and highlighted the similarity of the feeding structures (and possibly the feeding habits) between them, without any evidence of the presence of a suctorial oral disc in conodonts, admittedly it is not reasonable to assume these animals were ecological rivals in a given ecosystem.)

Line 326-332 – Interesting suggestion that the larger body size and hydromechanical advantage provided by fin morphology permitted upstream migration and the evolution of anadromy rather than vice versa. This is somewhat lost in what I think is still a longer, less focused discussion than it could/should be. **(Re: Thank you for your criticism. We have tried to revise and condense this part. Please check the new version of the MS. The implications of large body size and the hydromechanical advantages of the Jurassic lampreys were referred to in the first section of the ‘Discussion’. Maybe we should have made a simulation and conducted hydrodynamical analyses to show the functional significance of lampreys’ evolving body size and fin morphology, but it seems to be too long to include these in current paper)**

Reference 4 – It would be more useful if the specific chapter within this edited book is cited. It looks like some references to this source come from the introductory chapter (Docker et al. 2015, Introduction: A Surfeit of Lampreys, pp 1–34) while others (e.g., Line 302 come from Johnson et

al. 2015, Reproductive Ecology of Lampreys, pp 265–303) (**Re: Yes, indeed. Thank you for your suggestion. Now the pages of the specific cited chapters are marked in the ref. list according to the format of the journal (no title but the pages of the chapter).**

References 7, 14 – What are the title of these chapters? (**Re: Ref. 7 refers to “Chapter 3 Post-metamorphic Feeding in Lampreys”, according to the reference format of the journal, the pages of the cited chapters were marked in ref. list; Ref. 14 refers to “Chapter 4 Life History Evolution in Lampreys: Alternative Migratory and Feeding Types”**)